# Towards extending the aircraft flight envelope by mitigating transonic airfoil buffet

Esther Lagemann [1,2,3] ✉, Steven L. Brunton [1], Wolfgang Schröder [2] & Christian Lagemann [1,2,3]

In the age of globalization, commercial aviation plays a central role in maintaining our international connectivity by providing fast air transport services for passengers and freight. However, the upper limit of the aircraft flight envelope, i.e., its operational limit in the high-speed (transonic) regime, is usually fixed by the occurrence of transonic aeroelastic effects. These harmful structural vibrations are associated with an aerodynamic instability called transonic buffet. It refers to shock wave oscillations occurring on the aircraft wings, which induce unsteady aerodynamic loads acting on the wing structure. Since the structural response can cause severe structural damage endangering flight safety, the aviation industry is highly interested in suppressing transonic buffet to extend the flight envelope to higher aircraft speeds. In this contribution, we demonstrate experimentally that the application of porous trailing edges substantially attenuates the buffet phenomenon. Since porous trailing edges have the additional benefit of reducing acoustic aircraft emissions, they could prospectively provide faster air transport with reduced noise emissions.

Each aircraft has an individual flight envelope that ensures safe operations. It is typically determined during the design phase, as it defines operational limits with respect to aircraft speed, load factor, i.e., the ratio of lift to weight, and atmospheric conditions. Operations outside this envelope can result in severe structural damage, endangering flight safety and must be avoided at any time[1–3].

For modern lightweight aircraft operated in the transonic regime, the high-speed limit of the flight envelope is usually defined by the onset of aeroelastic stability problems[4,5]. The involved structural vibrations can constitute a severe threat to the structural integrity and a substantial body of research has focused on understanding the underlying phenomena[6–8]. The mechanisms of classical flutter are well understood and depend on the freestream dynamic pressure[5]. Flutter arises from two structural modes interacting via the flow, which enables a structural coupling with aeroelastic instability when reaching a particular flight speed, i.e., the critical flutter velocity. However, non-classical flutter occurring in transonic flows has different instability characteristics that also depend on the Mach number, i.e., the non-dimensional ratio of the local flow velocity to the local speed of sound, and the angle of attack, which refers to the angle between the freestream velocity and the inclination of the airfoil. Recent research has shown that transonic aeroelastic problems arise when the flow is close to or in an unsteady state[9–12] (for an extended literature review, the interested reader is referred to the Supplementary Information, section 3). This aerodynamic instability is called transonic buffet and describes self-sustained shock wave oscillations occurring on the aircraft wings[13–16]. Even though the aircraft is operated at subsonic speeds, i.e., the Mach number Ma is smaller than 1, the airfoil's shape causes an acceleration along the front part of its upper surface that creates a local supersonic region (Ma > 1). This region is terminated by

[1]AI Institute in Dynamic Systems, Department of Mechanical Engineering, University of Washington, Seattle, WA 98195, USA. [2]Institute of Aerodynamics, RWTH Aachen University, Wüllnerstraße 5a, 52062 Aachen, Germany. [3]These authors contributed equally: Esther Lagemann, Christian Lagemann. ✉e-mail: elage@uw.edu

a shock wave, which abruptly decelerates the flow to subsonic speed by an air compression. For particular flow conditions, which are defined by certain combinations of Mach number, Reynolds number, i.e., the ratio of inertial to viscous forces, and angle of attack, the shock wave becomes unsteady and oscillates in the streamwise direction[17,18]. Although extensive experimental and numerical investigations have been conducted to understand the underlying mechanism(s) of this aerodynamic phenomenon[1,19–27], its root causes and self-sustaining mechanisms are still controversially discussed[2]. Amongst the variety of hypotheses, one theory suggests that vortical structures propagate from the shock wave downstream to the trailing edge, where they generate acoustic waves that travel upstream and interact with the shock wave[28]. Various experimental measurements reported good agreement with this theory, e.g., refs. 20,21,29,30. Another hypothesis is based on the close relation between the shock oscillation and the time-dependent variation of the shock-induced separation, which essentially modifies the pressure ratio across the shock wave in a time-dependent manner[15,31,32]. However, despite discrepancies in the hypotheses about the underlying physical phenomena, it is unambiguous that transonic buffet constitutes a severe threat to aircraft operations. The self-sustained shock wave oscillations yield unsteady aerodynamic loads acting on the wing structure[11,33,34], and the aero-elastic structural response potentially interferes with the flight control system and/or could result in structural failure. Consequently, the respective flow conditions are excluded from the flight envelope. However, to enable faster and safe air transport, attenuating the aerodynamic instability associated with transonic buffet has the potential to tackle one of the limiting aeroelastic effects in the transonic flight regime and thus, is of substantial interest in aerospace engineering.

Therefore, it is not surprising that different strategies have been developed to optimize the high-speed aerodynamic performance by influencing the shock wave. Broadly, they can be divided into passive and active control methods. Passive approaches have the potential benefit of simplicity, robustness, low weight, and ease of retrofit[35]. While active methods are usually more complex and require a higher level of maintenance, their real-time feedback control is expected to provide better off-design performance compared to passive technologies.

A simple passive approach is to increase the structural damping. Since this comes at the cost of weight increase and thus, higher fuel consumption, it is certainly not the most efficient solution. Another passive approach is the installation of small bumps on the upper wing surface. These shock control bumps were shown to weaken shock waves, resulting in wave drag reduction, and to improve the buffet margin[35,36]. However, they are highly sensitive to the flow conditions, including shock strength, shock position, and post-shock pressure gradient[37]. An incorrect placement relative to the shock wave position results in the undesired effect of even stronger shock waves[38], which degrades off-design aerodynamic performance.

Another family of control devices, which comprises passive and active technologies, relies on the generation of vortices that energize the boundary layer via momentum transfer. The resulting stabilization of the boundary layer is directly linked to a shock wave stabilization, which was observed for mechanical, plasma, and fluidic vortex generators with pulsed and continuous blowing[39–41]. Although active devices possess the advantage of adjustability with respect to changing flow conditions via feedback control, the fixed location along the airfoil limits a flexibility with respect to off-design conditions for all vortex generators. Moreover, they increase drag[39,41], which degrades the aerodynamic performance.

To decouple the efficiency of control strategies from sensitive flow conditions, such as the shock wave position, research has focused on trailing edge devices. A prominent example are trailing edge deflectors and flaps, which are incorporated into a closed-loop control system that moves the devices in response to the shock wave. By counteracting the shock movement with trailing edge pressure changes induced by the flap movement, these devices can successfully control transonic buffet[42–46]. Their passive counterpart, i.e., a flap with a fixed deflection angle, is at least able to reduce the amplitude of the shock wave oscillations[47,48]. Such trailing-edge flaps are usually used as high-lift devices to improve the aerodynamic performance during take-off and landing. Consequently, a trade-off must be found with respect to a trailing-edge flap design that meets both needs—high-lift performance and buffet attenuation. Moreover, the deployment of high-lift components increases the far-field noise level, heavily contributing to the overall aircraft noise during landing[49]. However, due to the rising evidence of mental and physical health implications associated with aircraft noise[50–52], there is currently a strong demand for lowering acoustic emissions. In this respect, many academic studies have focused on trailing edge modifications since even for the clean wing, i.e., high-lift devices retracted, trailing edges constitute a dominant noise source term[49]. It has been shown in numerous studies[53–57] that porous trailing edges can efficiently reduce trailing edge noise.

In this contribution, we will demonstrate experimentally that porous trailing edges additionally constitute a promising technology for transonic airfoil buffet attenuation (see Fig. 1). This two-fold benefit is in great contrast to the other buffet control strategies discussed above. These devices typically increase the noise emissions because the additional components disturb the flow field[58,59]. Moreover, porous trailing edges have the advantages of low cost, low operational complexity, and robustness. Since they do not directly target the shock wave itself, the sensitivity to the flow conditions, and thus, an adverse off-design performance, is minimized. Moreover, these devices are easily manufactured using state-of-the-art 3D selective laser melting techniques, which allows an immense flexibility regarding design aspects such as the airfoil shape and material. Furthermore, we made two conscious design choices to minimize aerodynamic performance losses. First, lift degradation is counteracted by incorporating an impermeable plate inside the porous material. Without this blockage, a mass flux from the pressure to the suction side would yield a pressure compensation that has been shown to lower the aerodynamic lift[60–62]. Second, a perforated surface layer is added on top of the porous surfaces. This reduces the wall roughness and minimizes turbulence production, which was reported to increase the aerodynamic drag of porous surfaces[53,63,64].

A video research abstract of our work can be found at https://www.youtube.com/watch?v=ddomrz1cz_U&t=1s[65]. Overall, the findings presented in this contribution clearly demonstrate the ability of porous trailing edges to mitigate transonic airfoil buffet. Since one class of transonic aeroelastic instabilities of the wing structure is connected to the flow field unsteadiness associated with buffet[5,9,10,12], the aerodynamic stabilization is expected to positively impact the structural behavior for particular flow conditions. Thus, due to their benefits compared to other buffet control technologies, porous trailing edges possess great potential in advancing civil aviation by prospectively extending the limits of the flight envelope in the high-speed regime.

## Results

As a highly customizable and rapidly producible airfoil device, porous trailing edges present a quick, low-cost, and robust yet effective buffet control technique with significant practical advance for the entire aviation sector. To demonstrate experimentally the beneficial properties of porous trailing edges, comprehensive wind tunnel tests have been performed. We start by briefly outlining the experimental setup and the measurement facility. This is followed by a presentation of the flow field characteristics of the unmodified reference case, i.e., an ordinary solid trailing edge, before the buffet attenuation capabilities of our porous trailing edge designs are reported. Subsequently, we

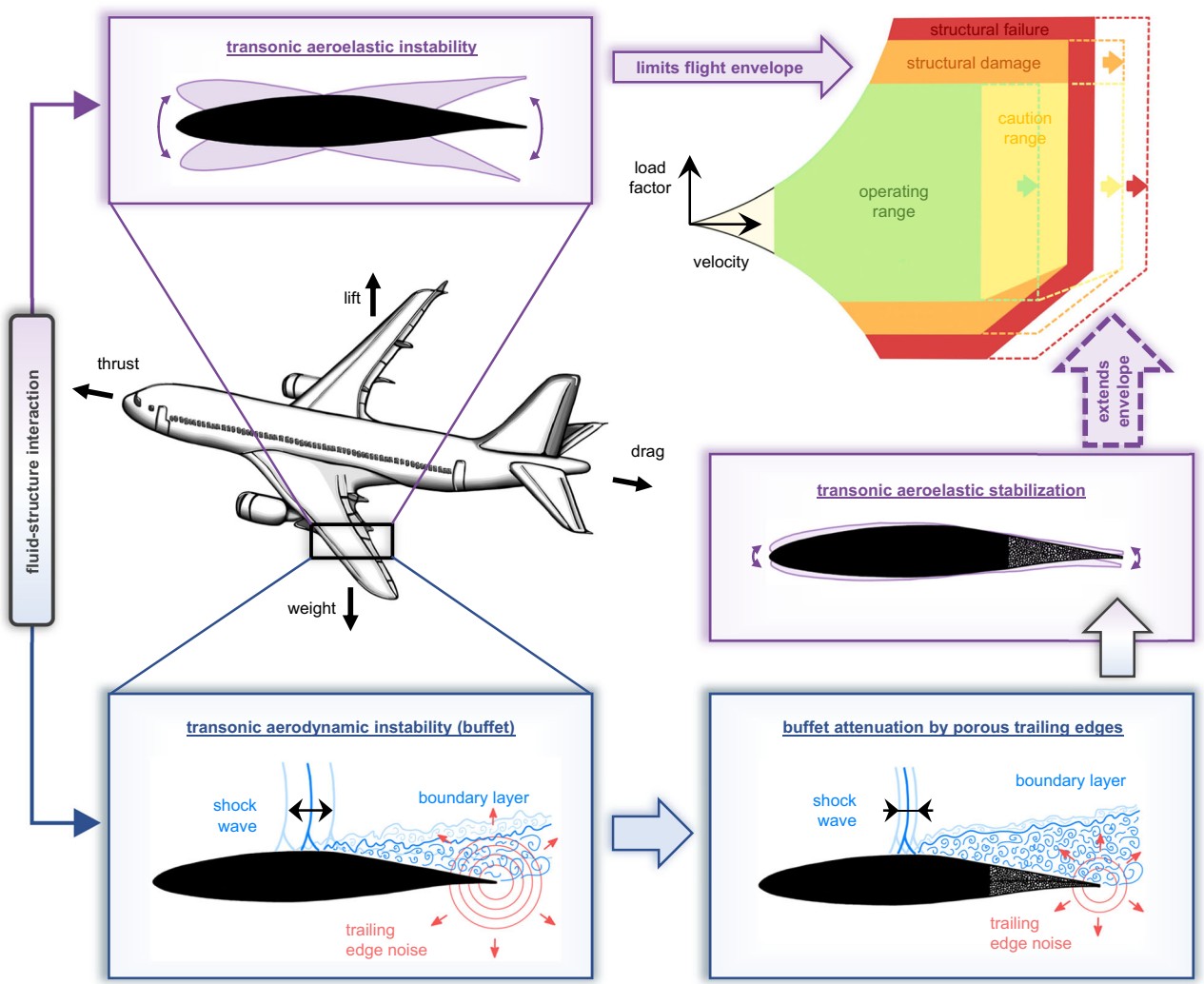

**Fig. 1 | Transonic buffet attenuation and flight envelope extension by porous trailing edges.** In the transonic flight regime, an aerodynamic instability called transonic buffet occurs at specific flow conditions. This phenomenon describes self-sustained shock wave oscillations along the suction side, i.e., the upper side, of the airfoil. Due to the fluid-structure interaction, certain aerodynamic and aeroelastic instabilities are coupled. Thus, the presence of transonic buffet is usually accompanied by structural instabilities, which potentially result in a structural failure of the aircraft wing. Therefore, the typical aircraft flight envelope is limited to lower aircraft speeds. We demonstrate experimentally that the installation of porous trailing edges attenuates the buffet phenomenon substantially. This allows an extension of the flight envelope to higher velocities without hazarding the aircraft's structural integrity. Such an extension is marked in the flight envelope diagram (upper right) by dashed lines and arrows. Moreover, porous trailing edges have been shown to reduce the trailing edge noise. Thus, our approach can lead to faster and safer aircraft operation with reduced noise emission.

discuss the aerodynamic performance of the investigated trailing edge designs. Finally, we elaborate on the physical mechanism by which the investigated porous trailing edges damp the self-sustained shock wave oscillations.

## Experimental setup

The experimental investigations are conducted in the Trisonic Wind Tunnel facility of the Institute of Aerodynamics at RWTH Aachen University. To investigate the desired flow conditions exhibiting distinct shock wave oscillations on the suction side, i.e., the upper side, of the airfoil, a Mach number of $Ma = 0.76$, an angle of attack of $\alpha = 3.5°$, and a chord-length based Reynolds number of $Re_c = 2.1 \cdot 10^6$ are chosen. To put it in perspective, this Reynolds number is at the lower bound of the spectrum for civil aircraft[66,67]. 

The rigid airfoil model's shape is based on the supercritical DRA 2303 profile, which was previously investigated in, e.g., refs. 6,20,21,29. The model possesses an exchangeable trailing edge section to study different configurations, i.e., a solid and two porous trailing edges.

The original solid trailing edge configuration is denoted as reference case (REF), whereas the porous trailing edges are abbreviated by PTE1 and PTE2. The PTEs consist of different deterministic structures, as depicted in Fig. 2. PTE1 is based on a three-dimensional lattice structure and PTE2 comprises stacked gyroid cubes. To prevent flow perturbations generated at sparsely connected struts, a perforated surface layer is added to both PTEs, which lines up precisely with the surface contour of the leading edge part. Moreover, a solid plate along the centerline of the profile prevents artificial mass flux and pressure compensation between the suction and pressure side of the airfoil to avoid aerodynamic performance degradation. Both porous trailing edges consist of titanium and are 3D-printed as a single part leveraging state-of-the-art selective laser melting processes.

To accurately resolve relevant quantities of the transonic airfoil flow in space and time, i.e., density gradient and velocity fields, a sophisticated multi-camera measurement setup is realized comprising simultaneous and synchronized planar high-speed Particle-Image Velocimetry (PIV) and Background-Oriented Schlieren (BOS)

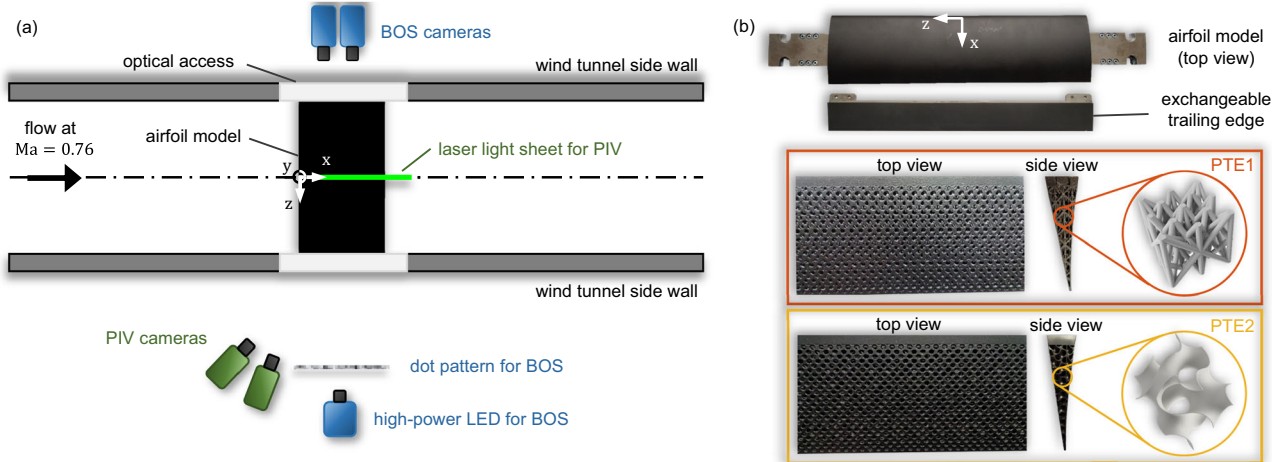

**Fig. 2 | Experimental setup. a** The experiments are conducted in the transonic measurement section equipped with a rigid airfoil model. The flow field around the airfoil is visualized by two synchronized measurement systems. The PIV setup captures the velocity field and the BOS setup measures density gradients, the latter allowing a precise localization of the oscillating shock wave. **b** Pictures of the airfoil model and the exchangeable trailing edges. Two types of porous trailing edges are tested: PTE1 is based on a three-dimensional lattice structure, and PTE2 comprises stacked gyroid cubes.

measurements (see Fig. 2). The PIV data provide streamwise and streamwise orthogonal velocity information while the BOS data contain density variations. Since a shock wave constitutes a discontinuous density jump in the flow field, its instantaneous position can be extracted precisely from the BOS data. The PIV data are used to understand how the PTEs influence the flow field in the trailing edge region to attenuate the buffet phenomenon. More details about the experimental setup and data evaluation are provided in the Methods section.

Throughout this paper, the following notation will be used: The $x$-coordinate and the streamwise velocity component $u$ point in the horizontal direction, and the $y$-coordinate and the normal velocity component $v$ are orthogonal to the $x$-coordinate as indicated in Fig. 2.

## Buffet characteristics of the reference case
Fully developed buffet characteristics describe a self-sustained shock wave oscillation along the suction side of the airfoil as sketched in Fig. 3a. The time-dependent shock wave position $x_s$ normalized by the chord length $c$ is shown in Fig. 3c. The typical range of motion is about 6% of the chord and, as depicted by the enlargement, it possesses periodic tendencies. Therefore, the buffet frequency peak, i.e., the prevailing frequency of the shock wave oscillation, can be extracted by applying a discrete Fourier transform to the time-dependent shock wave position. The corresponding normalized power spectrum is shown in Fig. 3b indicating a buffet frequency of $f \approx 180$ Hz or, when expressed in non-dimensional form, a Strouhal number of St $= fc/u_\infty \approx 0.109$ based on the chord length and the freestream velocity. This value is similar to previous investigations[21,29,68] with St $\approx 0.108$. The minor deviation can be explained by the different measurement techniques used to detect the shock wave movement. These former investigations used surface pressure probes, i.e., they considered the dynamics of the shock foot, while the present setup captures the entire shock wave using BOS measurements. Additionally, Fig. 3d depicts the probability density function (PDF) of the shock wave position, which comprises the same information as Fig. 3c but in a time-independent representation. Therefore, it provides easy access to the median shock position as well as its standard deviation. For this reference case, severe shock wave oscillations are observed with a standard deviation of std $(x_s/c) = 0.0159$. Please note that the previously mentioned studies[21,29,68] were conducted at a lower Mach number of Ma $= 0.73$. Therefore, the reported time-averaged shock wave position is located further downstream compared to the present case.

## Buffet attenuation effect of porous trailing edges
Next, the influence of two porous trailing edge designs consisting of different deterministic structures on the buffet characteristics is studied. Figure 4 shows the shock properties previously discussed for the reference configuration for all three test cases, i.e., REF, PTE1, and PTE2. The time-dependent shock wave position (Fig. 4a) clearly depicts the reduced shock wave oscillations induced by both PTEs. The time-independent representation provided by the corresponding PDFs in Fig. 4b convincingly shows narrower distributions with a stronger peak probability for both PTEs, which demonstrates the reduced shock wave oscillations. Moreover, the diagram reveals that the median shock position is moved slightly downstream, i.e., to higher $x_s/c$ values, when the PTEs are deployed. A quantitative comparison of the average shock position $\bar{x}_s/c$ and the corresponding standard deviation std($x_s/c$) is provided in Table 1.

In Fig. 4c-e, the normalized power spectra of the time-dependent shock position of all three configurations are shown. They clearly demonstrate that both PTEs substantially reduce the energy contained in the shock wave movement, i.e., they damp shock oscillations and eliminate the frequency peak usually associated with transonic airfoil buffet.

To understand how the PTEs interfere with the flow to attenuate the buffet phenomenon, we investigate the characteristics of the separated boundary layer downstream of the shock wave. For the studied flow conditions, the shock-induced boundary layer separation extends over the entire area between shock wave and trailing edge[69]. Due to the separation location, the PTEs can directly interact with the rear part of this boundary layer flow but not with the shock wave. As recently demonstrated by high-fidelity numerical simulations[15], the shock movement is directly coupled with the "breathing" of the boundary layer, i.e., a time-dependent thickening and thinning. Although these observations are made using a different airfoil geometry, we will show that the hypothesized mechanism is supported by the flow behavior of the DRA 2303 airfoil used in our experiments.

Figure 5a shows a PDF of the boundary layer thickness normalized by the chord length $\delta/c$. The boundary layer thickness is defined as the vertical distance to the airfoil surface at which the streamwise velocity reaches 99% of the freestream velocity. The values are extracted in the trailing edge part $x/c = [0.8, 1]$. When PTEs are deployed, the median boundary layer thickness is increased, which is revealed by a shift of the peak probability to larger values of $\delta/c$. Moreover, the time-dependent variation of the boundary layer thickness is reduced, which

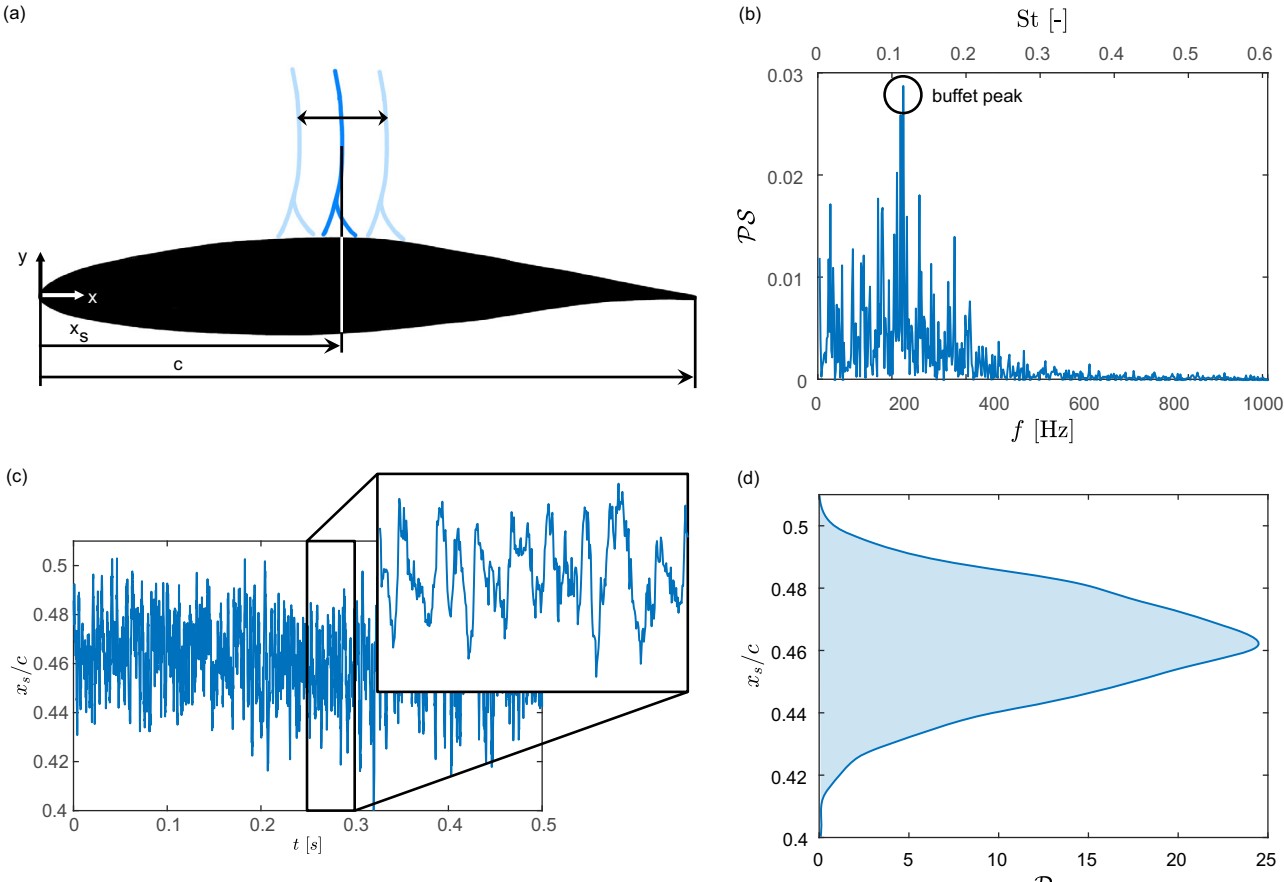

**Fig. 3 | Fully developed transonic airfoil buffet of the reference (REF) configuration.** A sketch of the shock wave oscillation along the airfoil is given in **a**, while (**b**–**d**) show measurement data related to the shock wave movement. The instantaneous shock wave position normalized by the chord length $x_s/c$ is given in **c** as a function of time $t$ and its PDF in **d**, where $\mathcal{D}$ denotes the probability density. The power spectrum $\mathcal{PS}$ of the shock wave position obtained by applying a Fourier transform is provided in **b** as a function of frequency $f$ and non-dimensional Strouhal number St. It reveals that the shock wave oscillates at a dominant buffet frequency of $f \approx 180$ Hz and St $= fc/u_\infty \approx 0.11$. Source data are provided as a Source Data file.

can be derived from the increased peak strength and the decreased standard deviation in the presence of the PTEs. Table 1 provides the respective quantitative values for these observations.

Figure 5 additionally depicts PDFs of the streamwise (b) and the vertical (c) velocity components within the boundary layer. It is obvious that both PTEs substantially change the velocity distribution in the boundary layer. The streamwise velocities (b) are shifted to smaller values, i.e., the area containing reversed flow enlarges. This means that the recirculation region is extended in the wall-normal direction. However, the velocity at the edge of the boundary layer reaches the same magnitude as in the reference configuration. The PDFs of the vertical velocity component (c) indicate a shift to larger velocity values in the presence of PTEs. To be precise, they are more symmetric around zero.

We noticed that the streamwise velocity distributions in the presence of PTEs resemble the distributions in pre-buffet conditions of the unmodified reference case. Therefore, Fig. 5d shows the respective PDFs related to the reference configuration for several Mach numbers. With the onset of buffet around Ma = 0.73, a re-distribution from low to high streamwise velocities is observed. The PDF's peak at velocities close to zero vanishes and only the high-velocity peak at the boundary layer edge remains. The pre-buffet distributions (Ma ≤0.72) with a double-sided peak are qualitative comparable to the PTE-related distributions at fully developed buffet, indicating the substantial importance of the streamwise velocity distribution inside the boundary layer for the buffet mechanism.

## Aerodynamic performance

The introduction of a technique for transonic airfoil buffet mitigation requires the investigation of its influence on the overall aerodynamic performance. In this respect, we made two conscious design choices, which are highlighted in Fig. 6, to minimize adverse effects on the aircraft's aerodynamics. First, we incorporate an impermeable titanium layer at the centerline, which prohibits any mass flux between pressure and suction side of the airfoil. Such a mass flux would result in a pressure compensation with adverse effects on the lift force. This performance degradation is extensively reported in studies investigating porous materials for noise reduction[60–62]. Second, we add perforated surface layers on both sides of the porous trailing edges, which are perfectly joined with the solid surfaces of the front airfoil part. Since previous investigations of porous trailing edges for noise reduction have noted a significant increase in turbulence intensity due to the roughness of the porous surfaces[53,63,64], we minimize this undesired effect by reducing the wall roughness substantially.

To ultimately prove the effectiveness of these design aspects, complementary measurements of lift and drag forces are required. Unfortunately, direct measurements with a high-precision force balance are not feasible since the wind tunnel facility does not offer such a measurement system for transonic flow conditions. Alternatively, the aerodynamic forces can be calculated by integrating the surface pressure along the airfoil. However, the incorporation of surface pressure sensors was discarded since a positioning within the porous

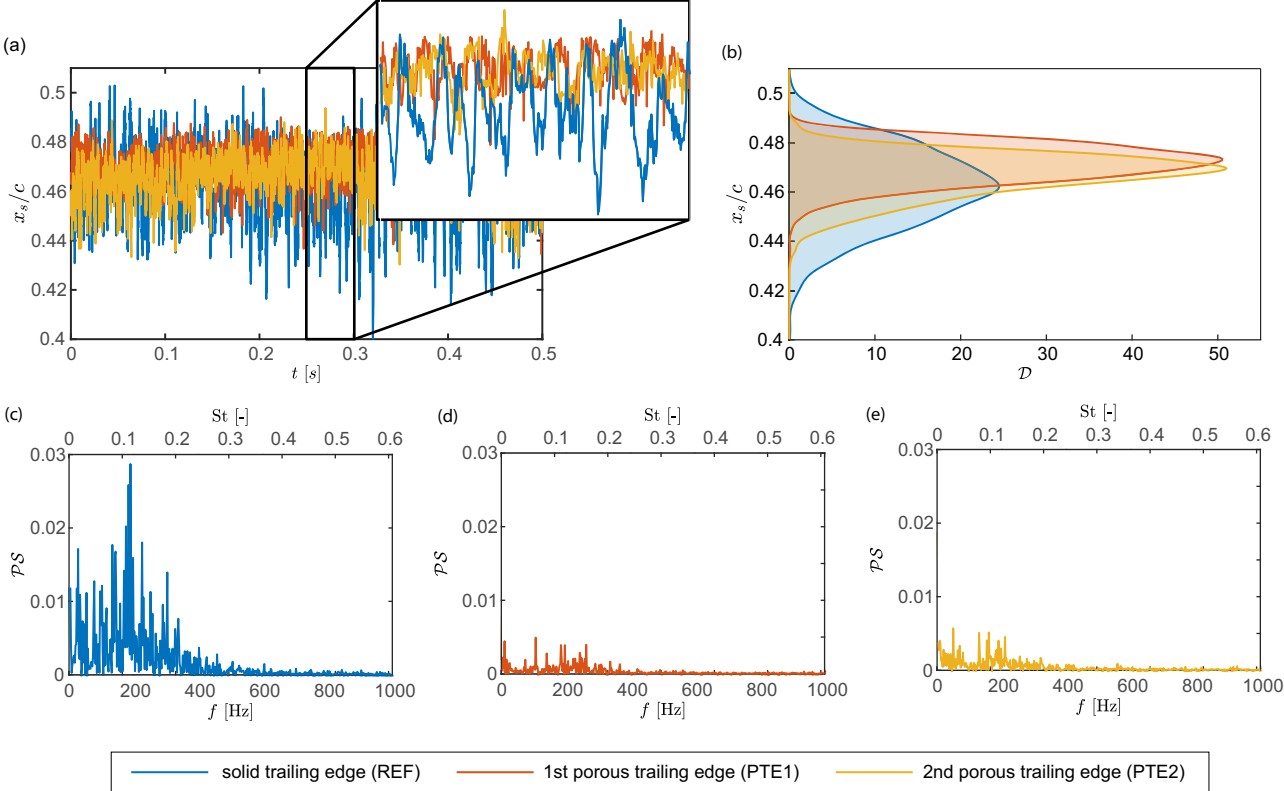

**Fig. 4 | Damping shock wave oscillations in fully developed buffet conditions using porous trailing edges.** The instantaneous shock wave position normalized by the chord length $x_s/c$ is given in **a** as a fucntion of time $t$ for both PTEs and the reference case (REF), while the corresponding PDFs are shown in **b**. Here, $\mathcal{D}$ denotes the probability density. Both representations convincingly demonstrate that both PTE designs successfully attenuate the shock wave movement. The power spectra $\mathcal{PS}$ of the shock wave positions obtained by applying a Fourier transform are provided in (**c**–**e**) as a function of frequency $f$ and non-dimensional Strouhal number St. They clearly reveal that both PTEs (**d, e**) substantially reduce the energy contained in the shock wave movement, i.e., they damp shock oscillations and eliminate the frequency peak usually associated with transonic airfoil buffet. Source data are provided as a Source Data file.

**Table 1 | Average ($\overline{(\cdot)}$) and standard deviation (std ($\cdot$)) of the shock wave position $x_s$ normalized by the chord length $c$ and of the boundary layer characteristics in the trailing edge part $x/c = [0.8, 1]$**

|      | $\overline{x_s}/c$ | std ($x_s/c$) | $\overline{\delta}/c$ | std ($\delta/c$) | $-\overline{u'v'}$ | $\overline{\lambda}$ |
|------|--------------------|---------------|-----------------------|------------------|--------------------|----------------------|
| REF  | 0.4620             | 0.0159        | 0.0970                | 0.0311           | 194.5              | 13787                |
| PTE1 | 0.4717 (2.1%↑)     | 0.0075 (52.8%↓) | 0.1069 (10.2%↑)     | 0.0238 (23.5%↓)  | 198.8 (2.2%↑)      | 13331 (3.3%↓)        |
| PTE2 | 0.4665 (1.0%↑)     | 0.0088 (44.7%↓) | 0.0988 (1.9%↑)      | 0.0208 (33.1%↓)  | 193.4 (0.6%↓)      | 13078 (5.1%↓)        |

These comprise the boundary layer thickness δ normalized by the chord length c, the median Reynolds shear stress $\overline{u'v'}$, and the strength of the vortical structures $\overline{\lambda}$. The latter is determined based on the swirling strength, which uses the imaginary part of the eigenvalue of the velocity gradient tensor to reveal rotational flow structures like vortices[94]. Values in parentheses depict the percentage difference to the values of the reference case.

trailing edges, i.e., the part where the most significant alteration of the aerodynamic forces is expected, was not possible without substantially disturbing the porosity of the material. Thus, the pressure field is derived from the conservation equations and the equations of state using the density gradient and the velocity fields from the BOS and PIV measurements. A comparison of the derived pressure distribution close to the wall with surface pressure sensor measurements of a previous study by Feldhusen et al.[68], which was conducted with the same reference airfoil in the same wind tunnel facility, is shown in Fig. 7(a). The data are provided at a slightly lower Mach number Ma = 0.73 since higher Mach numbers are not available from the literature. The satisfying agreement verifies the physical accuracy of our pressure calculation. Further details of this approach and the validation are described in the Methods section.

Figure 7b presents the pressure coefficient distributions of the reference configuration and both porous trailing edge designs at the Mach number of interest (Ma = 0.76). Reliable data are not available in the leading edge region and along the non-porous part of the pressure side due to optical constraints of the measurement section. However, the most significant changes of the flow field due to the porous trailing edges occur in the captured regions. Thus, the available information provides a reasonable trend of the aerodynamic performance changes in the presence of porous trailing edges.

By integrating the pressure distribution along the airfoil contour and distinguishing between the vertical (lift) and horizontal (drag) components of the resulting force vector, the contributions to the overall lift and drag values are calculated (details are provided in the Methods section). Figure 7c provides the lift and drag coefficients of all

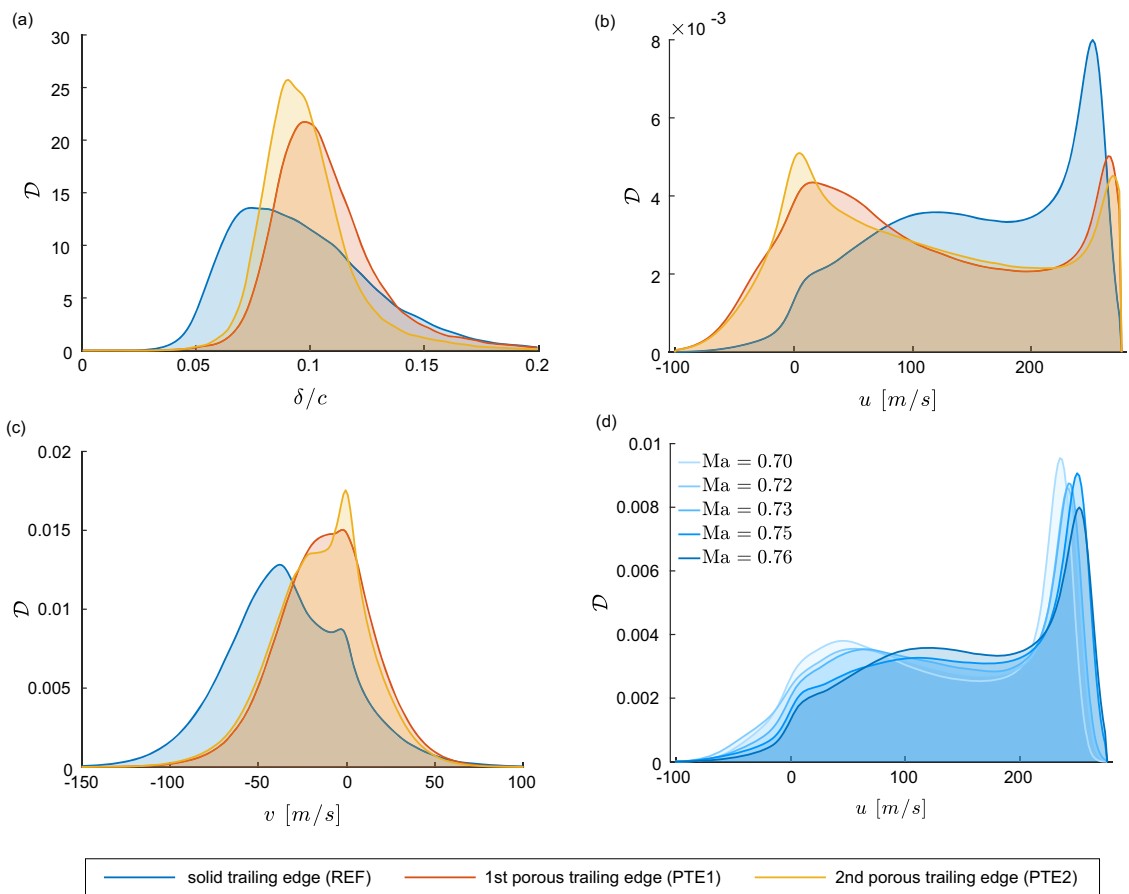

Fig. 5 | **Modification of the boundary layer characteristics in fully developed buffet conditions using porous trailing edges.** The PDFs, with $\mathcal{D}$ being the probability density, of the boundary layer thickness normalized by the chord length $\delta/c$ of both PTEs and the reference case (REF) are depicted in **a** and clearly reveal the reduced boundary layer breathing in the presence of PTEs. PDFs of the streamwise $u$ (**b**) and the vertical $v$ (**c**) velocity components within the boundary layer show the substantial modifications induced by the PTEs. A comparison to the streamwise velocity PDFs of the reference configuration at different Mach numbers Ma (**d**) reveals that the PTEs induce a streamwise velocity distribution similar to pre-buffet conditions (Ma ≤0.72). Source data are provided as a Source Data file.

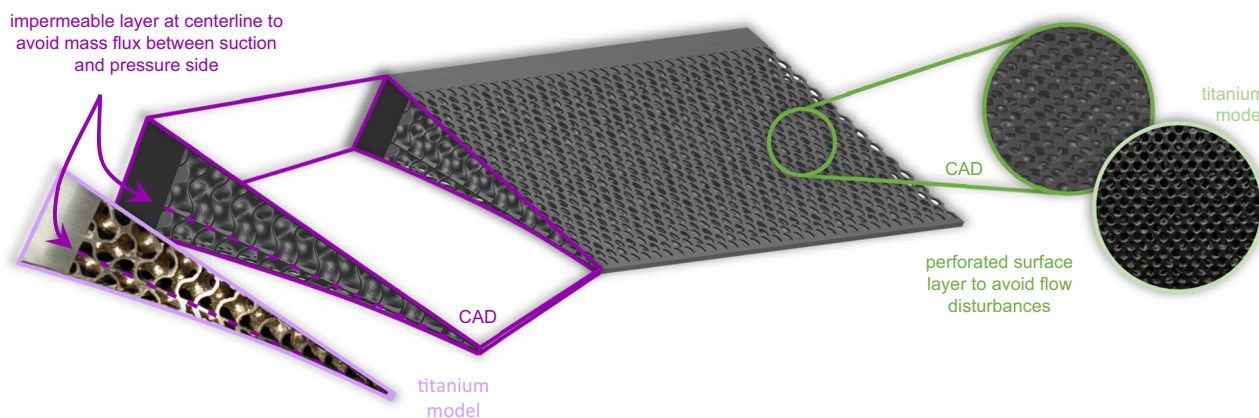

Fig. 6 | **Model design choices that optimize the aerodynamic performance of the porous trailing edge configurations.** When mitigating transonic airfoil buffet to extend the flight envelope in the high-speed regime, it is important to consider the effects of the respective airfoil modifications on the aerodynamic performance. To prevent a lift loss, an impermeable solid layer is added at the centerline. This blockage inhibits a mass flux between the pressure and suction side, which would yield a pressure compensation degrading the aircraft's lift force. Moreover, a perforated surface layer is added on both sides of the porous trailing edges. Since the porous material constitutes very rough surfaces, it typically disturbs the near-wall flow field massively resulting in increased turbulence intensity. This additional viscous drag would increase the overall aircraft's drag requiring a higher fuel consumption. Thus, both design choices aim at countering the aerodynamic performance penalties in the presence of porous trailing edges. The figure shows two representations of the PTE2 configuration, i.e., the Computer-Aided Design (CAD) model and photographs of the titanium model.

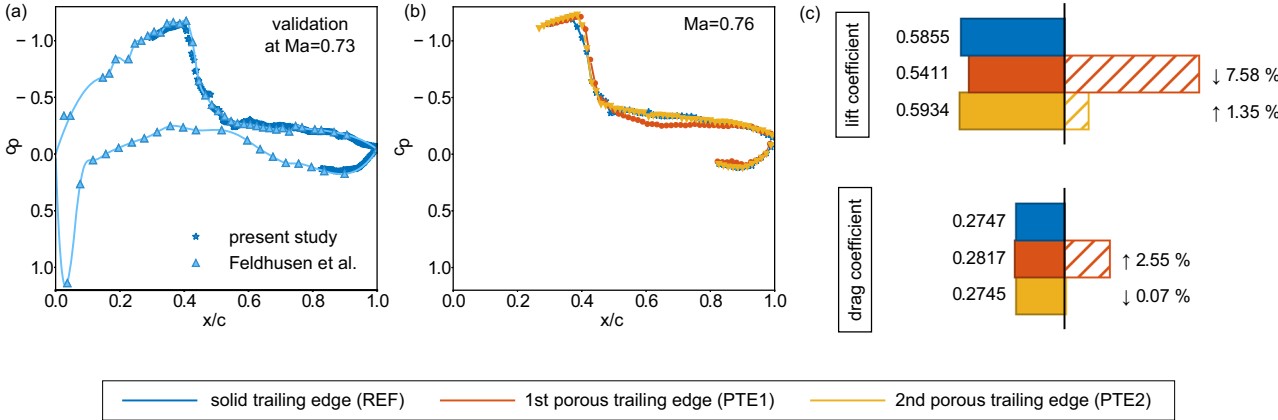

**Fig. 7 | Pressure distribution along the airfoil and lift and drag contributions.**
**a** time-averaged pressure coefficient distribution $c_p$ along the reference airfoil at Ma = 0.73 with the streamwise location $x$ being normalized by the chord length $c$. The derived pressure distribution is compared with surface pressure sensor measurements of a previous study by Feldhusen et al.[68] to verify the physical correctness of the pressure calculation. **b** time-averaged pressure coefficient distribution $c_p$ at Ma = 0.76 for all three configurations. Reliable data is not available in the leading edge region and along the non-porous part of the pressure side due to optical constraints of the measurement section. However, the porous trailing edges are not expected to change the flow field in the excluded regions significantly. A substantial deviation from the reference distribution is observed for PTE1 on the suction side. PTE2 follows the reference data quite well with a small deviation at the shock position. **c** Contributions to the overall lift and drag coefficients based on the pressure distributions given in **b**. The left-hand side displays physical values and the right-hand side provides the percentage deviation of the PTE configurations from the reference case. Arrows indicate if the deviation is positive or negative. PTE1 induces a strong reduction of the lift coefficient and an increase of the drag coefficient. On the contrary, PTE2 yields a small lift increase with a negligible variation of the drag coefficient. Source data are provided as a Source Data file.

three configurations and the respective percentage deviation from the reference flow when porous trailing edges are installed. The pressure distribution of PTE1 (b) and thus, the lift and drag contributions (c), show a significant deviation from the reference data. Precisely, a lift degradation of about 7.58% and a drag increase of around 2.55% are induced. On the contrary, the pressure distribution of PTE2 is similar to the reference case. The data indicate a small lift increase of 1.35% and a negligible variation of the drag coefficient of 0.07%.

The different aerodynamic performance of the porous trailing edge designs can be explained by the previous observations summarized in Table 1. The time-averaged boundary layer thickening of around 10.2% induced by PTE1 increases the aerodynamic drag, which was observed for other porous materials as well[58,60,64,70,71]. Furthermore, a minor increase of the Reynolds shear stress within the boundary layer of 2.2% could contribute to the aerodynamic performance deficit. Although the boundary layer breathing is substantially reduced by PTE1 (23.5% reduction of the boundary layer thickness variation), the stabilization by PTE2 is much higher (33.1%). Moreover, PTE2 does not increase the Reynolds shear stress and yields only a small boundary layer thickening of 1.9%. We conclude that these effects result in a negligible variation of the drag coefficient for PTE2. The minor lift increase can probably be traced back to the small downstream shift of the time-averaged shock wave location.

Using a phenomenological explanation, the diverging effect of the investigated porous trailing edge designs arises from the distinct difference of the porous materials. PTE1 is made of a well-ordered lattice structure, in which it is easier for the fluid to find a way out and back into the main flow. The stacked gyroids of PTE2 create an intricate and labyrinth-like structure such that it is more likely for the fluid to get "trapped". Therefore, PTE2 is still able to extract kinetic energy from the main flow and damps oscillations but it does not "release" as much low momentum fluid back into the outer flow compared to PTE1. This reduced fluid exchange between porous structure and outer flow compared to PTE1 yields the observed variations of the boundary layer properties, i.e., less boundary layer thickening, less boundary layer breathing, and no increase of the turbulence stress.

In conclusion, although both porous trailing edge designs are able to mitigate transonic buffet, their impact on the aerodynamic performance varies substantially. A performance degradation relative to both, lift and drag forces, is observed for PTE1, whereas PTE2 influences the aerodynamic behavior only slightly but in a favorable manner. Hence, this design counteracts the aerodynamic penalties previously reported in the literature for porous materials, which establishes a cornerstone for prospective research and real-world applications.

## Physical mechanism underlying the buffet mitigation effect of porous trailing edges

The experimental studies demonstrate that both investigated porous trailing edge designs substantially attenuate the adverse shock wave oscillations associated with transonic airfoil buffet. In the following, we will explain how the PTEs interfere with the flow field to mitigate the shock wave movement since their location does not allow a direct interaction with the shock wave itself. The explanation is illustrated by the schematic diagram given in Fig. 8. Essentially, the success of PTEs is rooted in the physical coupling of the shock wave oscillations and the boundary layer thickness variation, the latter being directly influenced by the PTEs.

When replacing a solid surface by a porous material, fluid penetrates the porous structure. This is known as the transpiration effect, which effectively reduces the kinetic energy of the fluid flow in the vicinity of the porous surface. In the present scenario, this effect can be observed based on the altered velocity distributions in the boundary layer (Fig. 5b,c). The streamwise velocity data additionally reveal that this effect causes a thickening of the recirculation region, i.e., an increased mass flux of upstream moving fluid, which results in a thicker boundary layer (Fig. 5a).

A thicker boundary layer possesses an increased resistance to perturbations. Incoming disturbances are more efficiently damped, which reduces the boundary layer breathing as depicted in Fig. 5a and Table 1. In turn, the attenuated upwards and downwards movement of the boundary layer stabilizes (i.e. reduces) the pressure fluctuations above the rear part of the airfoil. Since such pressure fluctuations directly affect the shock wave position, their damping results in a

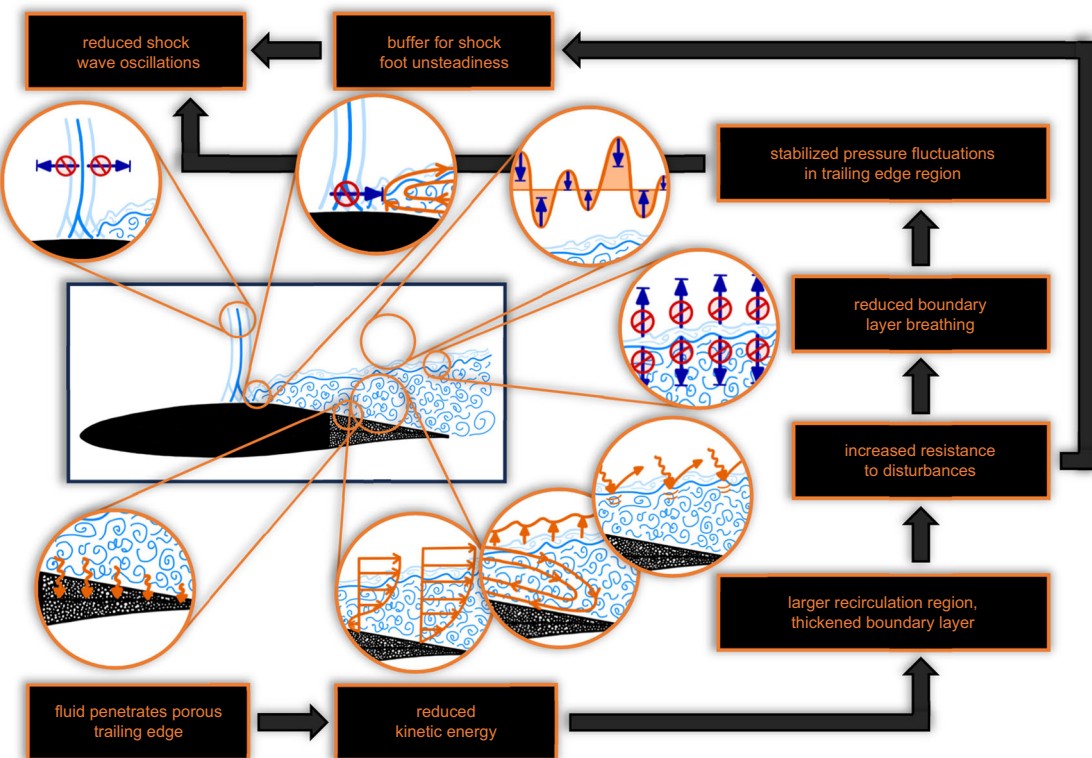

**Fig. 8 | Schematic diagram explaining how porous trailing edges interact with the flow field to mitigate the shock wave oscillations of transonic airfoil buffet.** In the presence of PTEs, fluid penetrates the porous material, which results in a reduction of the kinetic energy of the flow. Consequently, the recirculation region enlarges and the boundary layer thickness increases. Therefore, the boundary layer has an increased resistance to incoming disturbances, which provokes two different effects. First, the boundary layer breathing is reduced, which damps the pressure fluctuations in the trailing edge region. Second, the recirculation region acts as a buffer at the shock foot, i.e., it damps instabilities associated with the shock wave. Both effects cause an overall reduction of the shock wave oscillations. Consequently, the physical connection between the boundary layer downstream of the shock wave and the shock wave itself enables a mitigation of the shock wave movement via porous trailing edges, although the location of such trailing edge devices inhibits a direct interaction with the shock wave. Please note that the physical processes sketched in this figure are simplified and exaggerated to illustrate the respective phenomena more clearly.

reduction of the shock wave movement. As a second contributor, the extended recirculation region behind the shock wave (Fig. 5b) stabilizes the shock wave by reducing the unsteadiness associated with the interaction region. It basically acts as a buffer absorbing perturbations that would otherwise provoke a shock wave movement. These connected phenomena explain why porous trailing edges are effective devices for transonic buffet attenuation, although their location does not allow a direct interaction with the shock wave. This explanation is supported by studies from the literature, which identify the shock foot instability and the connection between shock wave oscillation and boundary layer breathing as the driving mechanisms for transonic airfoil buffet, e.g., refs. 15,22,26,31,32.

In addition to these observations, we noticed that the streamwise velocity distribution inside the boundary layer in the presence of PTEs is comparable to the distribution in a pre-buffet configuration (Fig. 5d, Ma ≤0.72). However, the averaged shock position, the averaged boundary layer thickness, and the vertical velocity distribution inside the boundary layer differ from the pre-buffet conditions. Thus, the streamwise velocity distribution inside the boundary layer is an essential feature for transonic airfoil buffet. Consequently, its modification can attenuate the buffet phenomenon due to the direct coupling between the boundary layer properties and the shock wave movement as described above.

Finally, the differences between the porous trailing edge designs, PTE1 and PTE2, are addressed. The thickened boundary layer in the presence of PTEs effectively changes the surface contour imposing a favorable pressure gradient. This means that the pressure in the trailing edge region is reduced, which shifts the average shock position downstream, as observed in Fig. 4 and Table 1. Since the boundary layer thickening of PTE2 is less intense, the shock position downstream shift is also less severe. The varying damping efficiency of both PTEs regarding the boundary layer breathing can be traced back to the different porous structures of the PTEs. Therefore, PTE2 reduces the fluctuation of the boundary layer thickness more efficiently compared to PTE1.

## Discussion

Porous trailing edges have attracted substantial interest in recent years since they lower acoustic aircraft emissions[71,72]. The present study reveals a second benefit of PTEs - their ability to attenuate self-sustained shock wave oscillations associated with the transonic buffet phenomenon. Several recent studies observed a direct connection between this aerodynamic instability and a particular class of transonic aeroelastic stability problems[9–12] (a review of recent literature is provided in the Supplementary Information, section 3), which constitute a hazard to flight safety due to their undesirable impact on the flight control system and the structural integrity. Therefore, the present findings are important for the aviation sector in outlining a promising strategy to mitigate certain safety-critical aerodynamic and aeroelastic instabilities. This could allow to safely extend the high-speed limit of the flight envelope while simultaneously reducing aircraft noise. Moreover, the results of this study provide insight into the physical mechanism underlying the buffet phenomenon. This lays the foundation for prospective aircraft

designs in which the PTEs can be precisely tailored to the expected flow conditions.

In summary, our experimental study shows that PTEs modify the velocity distribution within the separated boundary layer downstream of the shock wave. Since the altered streamwise velocity mimics the characteristics of pre-buffet flow conditions, its modification via the PTEs has an essential impact on the shock wave oscillations. The extended recirculation region damps instabilities associated with the shock-wave/boundary-layer interaction, while the less intense boundary layer breathing reduces pressure fluctuations in the trailing edge region with a direct impact on the shock wave oscillation due to the inherent connection between the shock wave position and the downstream pressure. In other words, a reduction of downstream pressure variations stabilizes the shock wave and damps transonic buffet.

The functional structure of the porous material has a distinct impact on the aerodynamic performance of the modified airfoil. Although both investigated designs successfully mitigate transonic buffet, a significant lift and drag degradation was observed for the lattice-based trailing edge (PTE1), while a porous structure based on gyroid components (PTE2) influences the aerodynamic forces only slightly but favorably. Thus, a careful trailing edge design with an adequate choice of the porous material is a key element for future aircraft applications.

The present investigation does not allow comprehensive conclusions about the maximum efficiency of PTEs in the context of buffet mitigation since only two PTE designs have been tested under fully developed buffet conditions. However, the manufacturing of PTEs via selective laser melting processes allows a flexible and fast design facilitating prospective parameter studies with different porous structures. Moreover, it is interesting to assess how PTEs influence the buffet onset and offset boundaries, i.e., how far they can shift the occurrence of shock wave instabilities to higher Mach numbers and/or angles of attack. Complementary high-fidelity numerical simulations could provide time-resolved lift and drag values for the entire airfoil although an adequate implementation of a porous trailing edge at these flow conditions is highly challenging. Alternatively, additional experiments with a force balance or surface pressure measurements beyond the porous material could be conducted. It might also be beneficial to incorporate the benefits of machine learning based data assimilation techniques leveraging the power of latent dynamical models[73] to extract physical mechanisms or causal learning frameworks[74] to derive the inherent causal relations.

Finally, we like to point out that the present study focuses on the aerodynamics of 2D transonic airfoil buffet and neglects aeroelasticity as well as 3D effects occurring on realistic swept wings where the nature of the buffet phenomenon can be additionally influenced by interactions with, e.g., the engine nacelle[75] and the aircraft fuselage[76]. However, the dominant aerodynamic phenomena of 3D straight and mildly swept wings are similar to 2D airfoil buffet[12,77,78]. Therefore, the present findings provide a good indication of the behavior of the complex real-world system and thus, a valuable basis for more sophisticated prospective investigations, which certainly have to be conducted prior to approaching real-world aircraft implementations. Especially with respect to the efficiency of porous trailing edges, prospective studies should involve aeroelastic considerations to contribute to the on-going research question to which extent transonic buffet and structural stability issues are related.

## Methods
### Experimental setup
The experimental measurements are conducted in the Trisonic Wind Tunnel facility at the Institute of Aerodynamics of RWTH Aachen University, which operates at subsonic, transonic, and supersonic conditions. The present investigations are conducted within the transonic measurement section at Mach numbers in the range of Ma = {0.70; 0.76}. The main body of research focuses on measurements at Ma = 0.76 since these flow conditions exhibit fully developed buffet conditions. Only the reference configuration with an ordinary solid trailing edge is additionally investigated at lower Mach numbers. These data are compared with the modified flow topology induced by the porous trailing edges to assess if these devices transfer the flow field into a pre-buffet condition.

The experimental airfoil model is based on a supercritical DRA 2303 profile, which was extensively investigated in former experimental and numerical studies[6,20,21,29]. It has a relative thickness-to-chord ratio of 14% with a total chord length of $c = 150$ mm. The leading edge part ($0.7c$) and the solid trailing edge are made of stainless steel, while the porous trailing edges are made of titanium. A zigzag strip is glued onto the airfoil model at about 5% chord to trigger a boundary layer transition. This is necessary to simulate a turbulent boundary layer flow similar to real flight conditions.

The airfoil model is installed in the measurement section at a fixed angle of attack of $\alpha = 3.5°$ relative to the freestream velocity. The Reynolds number based on the freestream velocity and the chord length of the airfoil is $Re_c = 2.1 \cdot 10^6$. Since the wind tunnel is operated at ambient flow conditions, Reynolds number and Mach number are linked via the fluid state and cannot be adjusted separately.

To exclude any influence of the wind tunnel walls on the flow field around the airfoil model, the measurement section has adjustable side walls. They are calibrated based on 26 dynamic pressure transducers distributed along the centerline of the upper and lower wall such that the wall curvature follows exactly the streamlines, which simulates an unbounded environment like in real flight conditions[21].

The wind tunnel itself is an intermittently working vacuum storage facility in which the fluid flow is generated by the pressure difference between a vacuum tank and an air-filled balloon[21]. Prior to each measurement run, the reservoir tank downstream of the measurement section is evacuated, while the dehumidified air is stored in the balloon upstream of the measurement section acting as a stagnation chamber. A silica gel-based drier ensures a relative humidity of the air below 4% at a total temperature of about 293 K[20] such that no condensed water impedes the measurements. By opening a shutter, the pressure difference between the evacuated tank and the balloon creates an airflow from the balloon into the tank through the measurement section. Thereby, stable measurement conditions are present for about 2 seconds in each measurement run[21]. Due to the limited storage capacity of the high-speed cameras, the duration of one measurement is limited to about 85 buffet cycles.

To simultaneously capture the shock wave motion and the fluid flow in the trailing edge region, synchronized Particle-Image Velocimetry (PIV) and Background-Oriented Schlieren (BOS) measurements are conducted. The BOS data are acquired with two Photron Fastcam SA5 cameras equipped with 180 mm Tamron f/8 lenses. High-power LEDs illuminate a random dot pattern, which is recorded by the cameras at a frame rate of 8000 Hz. Any density gradient changes in the flow field create an optical distortion of the dot pattern, which is captured in the BOS images. Since a shock wave constitutes a discontinuous density jump, the shock wave oscillation can be precisely studied via the BOS setup.

To capture the velocity field, simultaneous PIV measurements are conducted. Therefore, Di-Ethyl-Hexyl-Sebacat (DEHS) particles are added to the airflow directly within the balloon. The particles' properties are matched to the flow properties such that they perfectly follow the fluid flow. They are illuminated by a pulsed Darwin Duo 527-100-M laser at 4000 Hz and captured by two pco.dimax HS4 high-speed cameras equipped with 100 mm Makro-Planar T* ZF.2 lenses. Due to limited optical access, the PIV cameras are setup at an angle relative to the laser light sheet (see Fig. 2). Scheimpflug adapters are used to compensate this shift in perspective. Examples of the raw PIV and BOS images are provided in the Supplementary Information in

section 1 in Fig. S1, while Fig. S2 provides the corresponding instantaneous velocity and density gradient fields.

The porous trailing edges are manufactured using selective laser melting processes, which is a 3D printing technique based on melting and recombining metallic powder using a high-power laser. The two porous trailing edges used in the present study differ in the functional structure and permeability. The first trailing edge (PTE1) has a latticed structure, and the second device (PTE2) consists of stacked gyroid cubes. Its composition makes it mechanically more stable, which is an important factor for prospective real-world applications. To prevent undesired flow perturbations at sparsely connected struts close to the trailing edges' surface, a perforated surface layer is added to both PTEs as shown in Fig. 6. Furthermore, a solid plate is included along the centerline of the trailing edge profile to inhibit a mass flux and thus, a pressure compensation, between the suction and the pressure side of the airfoil, which would degrade the aerodynamic performance.

### Data evaluation

A state-of-the-art in-house code[79,80] is used to evaluate the PIV data. The data consist of double images, i.e., two consecutive images acquired within a short time frame, which are used to estimate the particles' displacement in a statistical sense via cross-correlation. The known time interval between the two images is then used to transfer the horizontal and vertical displacement field into horizontal and vertical velocity information. The whole evaluation procedure includes sophisticated state-of-the-art processing tools to achieve the highest physical accuracy of the resulting velocity fields. That is, a multigrid approach with steps $8 - 4 - 2$ is deployed, where the size of the interrogation windows is progressively reduced until the final window size $16 \times 16$ px$^2$ with an overlap of 75% is reached. A five-step predictor-corrector scheme incorporated into an iterative process allows for the highest accuracy in the displacement estimation. A Gaussian peak fit estimator enables sub-pixel accurate estimates, while a normalized median test reliably detects outliers. More details about the evaluation scheme are provided in refs. 79,80.

The BOS data are evaluated with a deep optical flow network called RAFT-PIV[81–83], which was recently proven to provide accurate density gradient estimates from BOS data[84]. In principle, each BOS image is compared to the initial reference image at zero flow to extract the changes in the density gradient due to the fluid flow. Compared to established evaluation routines, RAFT-PIV provides a flow state estimate for each pixel. This massively increases the accuracy and the spatial resolution of the final output compared to cross-correlation based methods. It is especially helpful in the present context to precisely extract the shock position. Since the data are not averaged across the finite size evaluation window like in classical approaches, the shock localization is significantly more precise. Moreover, we further enhance the shock detection by applying the 2D Noise-Assisted Multivariate Empirical Mode Decomposition (2D NA-MEMD)[85] to the density gradient fields provided by RAFT-PIV. This data-driven method extracts scale-based modal representations from the input data in a spatio-temporal framework based on the scales inherent to the data. The sharp edges of the shock wave distinctly appear in the first mode, which allows an accurate extraction of the shock wave position for each individual time instant. A temporal coherence is achieved by simultaneously decomposing consecutive snapshots of the flow field. This temporal consistency is further enhanced by deploying the concept of noise assistance, where we use two additional channels containing random Gaussian noise with a standard deviation of 2% of the standard deviation of the multivariate data in the decomposition.

The reader is referred to refs. 84–87 for further details on the 2D NA-MEMD and its application to fluid flow data.

### Calculation of the aerodynamic quantities

The wind tunnel facility is neither equipped with a force measurement system nor have surface pressure sensors been incorporated into the airfoil model due to constraints with respect to the porous trailing edges. Therefore, the aerodynamic performance of the airfoil is derived from the flow field measurements. Once the pressure field is known, the surface pressure can be integrated along the airfoil contour to provide a reasonable approximation for the lift and drag values. The pressure field is derived in three steps (see Fig. 9). First, the density gradient fields extracted via the BOS measurements are integrated with a weighted least-squares optimization methodology[88] yielding the density field (Fig. 9a). Second, the temperature field is determined using the velocity data from the PIV measurements in combination with the energy equation (Fig. 9b). Third, the ideal gas law is applied to calculate the pressure field from the density and the temperature field (Fig. 9c). In contrast to other studies estimating the pressure field and the aerodynamic performance from PIV data of the compressible flow around an airfoil[89,90], we additionally have the unique possibility to invoke the density information from the BOS measurements. This allows the derivation of the quantities of interest with a higher accuracy, involving less assumptions about the flow state.

Since the BOS and the PIV data have a different spatial resolution, the BOS-based flow quantities are interpolated onto the same grid as the PIV data. To calculate the density, temperature, and pressure fields, we use the time-averaged velocity and density gradient fields. Since our approach relies on a few assumptions, e.g., ideal gas properties and isentropic flow upstream and downstream of the shock wave, respectively, we expect our results to be physically significant in a statistical sense. In fact, we are able to verify the accuracy of the time-averaged pressure data calculated for the reference configuration, i.e., with an ordinary trailing edge, at a Mach number of Ma = 0.73 and an angle of attack of $\alpha = 3.5°$ with pressure tap measurements[68], which have been conducted for the same airfoil in the same wind tunnel facility. The good agreement, which can be observed in Fig. 9d, shows that our approach provides physically meaningful pressure information.

The derivation of the pressure fields and the validation based on previous surface pressure measurements is outlined in greater detail in the Supplementary Information in Section 2.

The aerodynamic force and thus, the lift and drag values, are obtained by integrating the surface pressure along the airfoil. Due to the two-dimensional nature of the airfoil, the force $F$ is calculated per unit depth such that the area used for the pressure integration reduces to the airfoil perimeter $S$ (equation (1)). Since we only have discrete pressure values, we use a Riemann sum to approximate the integral. In this case, the local surface increment ds$_i$ is tangential to the airfoil surface since the pressure acts perpendicular to the surface. The increment ds$_i$ is obtained from a piecewise linear interpolation of the profile coordinates. To obtain the horizontal and the vertical force components, i.e., the drag $D$ and lift $L$ contributions, respectively, we apply equations (2), (3). The angle $\alpha$ is location-dependent and can directly be determined from the profile coordinates (again, using a piecewise interpolation). The surface normal vector $\vec{n} = \left( n_x, n_y \right)$ accounts for the sign of the respective quantity. Subsequently, all contributions are summed up to yield the overall drag and lift values. Based on equation (4), the lift and drag coefficients are calculated using the freestream quantities.

Please note that viscous stresses are neglected for the calculation of the lift and drag since their contribution is negligibly small

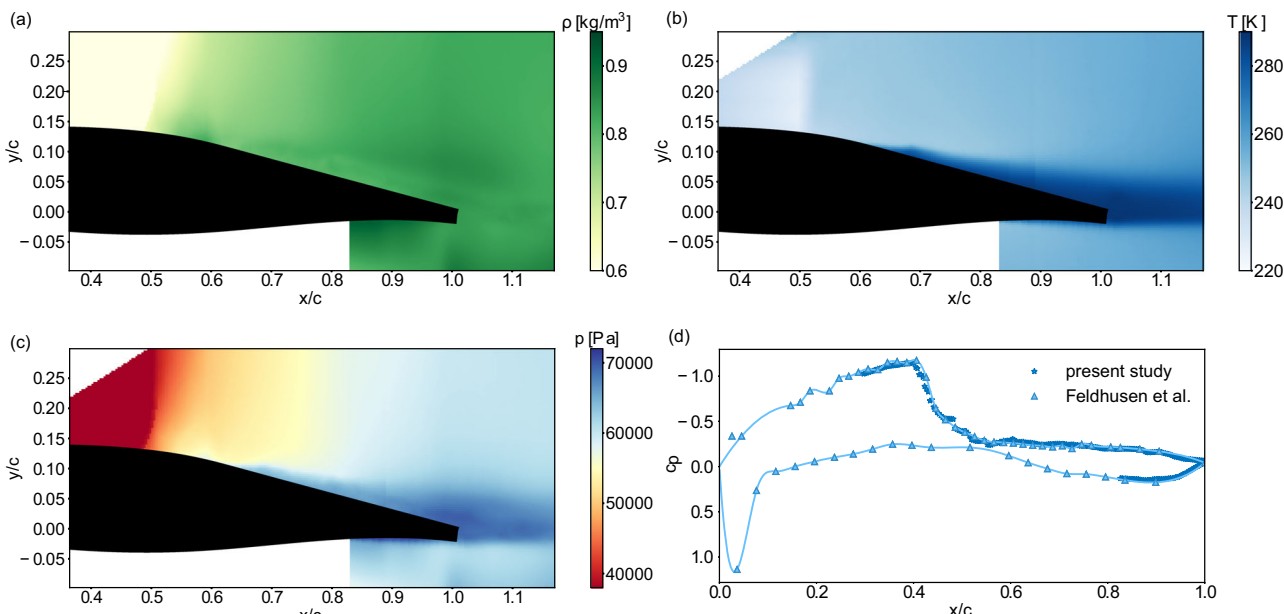

**Fig. 9 | Pressure field derivation to calculate aerodynamic performance measures based on the time-averaged flow field information at Ma = 0.73.** An integration of the density gradient fields provides the density field $\rho$ as a function of streamwise $x$ and vertical $y$ location as shown in **a**. The temperature field $T$ given in **b** is derived from the velocity data and the energy equation. Finally, the ideal gas law is used to provide the pressure field $p$ shown in **c**. A comparison of the pressure distribution close to the wall with surface pressure measurements conducted by Feldhusen et al.[68] in **d** shows that the present approach is able to derive physically meaningful pressure coefficient values $c_p$, which is of utmost importance for the subsequent calculation of the aerodynamic quantities. Source data are provided as a Source Data file.

compared to the pressure effects[89].

$$F = \int_S p \, d\vec{s} \approx \sum_S F_i = \sum_S p_i \, ds_i \qquad (1)$$

$$D = \sum_S F_i \cdot \sin \alpha_i \cdot n_{x,i} \qquad (2)$$

$$L = \sum_S F_i \cdot \cos \alpha_i \cdot n_{y,i} \qquad (3)$$

$$c_f = \frac{f}{0.5 \rho_\infty u_\infty^2 S} \quad \text{with } f = L, D \qquad (4)$$

Since our BOS and PIV measurements do not capture the entire airfoil due to accessibility constraints of the measurement section (see Fig. 9), we cannot calculate the overall lift and drag values since we do not have pressure information at the entire airfoil surface. However, we have pressure data in the regions where the highest alteration due to the porous trailing edges is expected. That is, around and downstream of the shock wave on the suction side and along the porous trailing edge on the pressure side. Therefore, we calculate how the contributions of these regions to the overall drag and lift values change in the presence of porous trailing edges. This convincingly shows the governing trends of the aerodynamic performance. As a reference, the complete pressure data provided in ref. 68 is used to estimate drag and lift values for the entire airfoil. Subsequently, the lift and drag coefficients based on the restricted area of the present study are calculated and compared between a pressure tap-based and a flow field-based analysis, which is discussed in greater detail in the Supplementary Information in section 2.7.

## Data availability

Source data are provided with this paper.

## Code availability

Source code of the 2D NA-MEMD is available at ref. 91 and of RAFT-PIV at ref. 92. The density integration methodology is provided at ref. 93. Standard built-in functions of Matlab and Python are applied to calculate the FFTs and PDFs.

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

## Acknowledgements

The authors like to thank Julia Roeb, Julian Knöll, and Julian Brühl for assisting in the measurement campaign and data processing as well as the whole workshop team from the Institute of Aerodynamics for their technical support. This research was funded by the German Research Foundation within the Walter Benjamin fellowships MA 10764/1-1 (EL) and LA 5508/1-1 (CL) and the experimental measurement campaign within research grant KL 2138/5-1 (WS). EL, CL, and SLB acknowledge support from the National Science Foundation AI Institute in Dynamic Systems (grant number 2112085) and the Boeing Company. Furthermore, the authors gratefully acknowledge the Gauss Centre for Supercomputing e.V. for funding this project by providing computing time on the GCS Supercomputers (CL).

## Author contributions

Conceptualization: E.L., S.L.B., W.S., C.L. Investigation: E.L., C.L. Visualization: E.L. Supervision: C.L. Writing-original draft: E.L., C.L. Writing-review & editing: E.L., S.L..B., W.S., C.L.

## Competing interests
The authors declare no competing interests.
