## [Peer Review File · Nature Communications]

Towards extending the aircraft flight envelope by mitigating transonic airfoil buffetReviewers' comments:

Reviewer #1 (Remarks to the Author):

The study examines the DRA2303 airfoil in buffet conditions (Mach 0.76, 3.5° angle of attack). Two porous trailing edges (TE) are fitted to the airfoil, demonstrating attenuation in shock oscillations. To my knowledge, this research is original as previous studies on shock buffet mitigation did not explore TE adjustments like those proposed in this study.

The authors link the reduction in shock buffet oscillations to the possible expansion of the flight envelope. Although aerodynamic forces due to shock buffet might induce structural loads that prohibit flight at buffet conditions, the flight envelope at high dynamic pressures is constrained by flutter, not buffet. Figure 1 is, therefore, misleading in relating buffet reduction to expanding the flight envelope. Furthermore, the study focuses on an airfoil buffet, a distinct mechanism from a wing buffet. Therefore, the assertion that the suggested TE devices can expand the flight envelope lacks substantiation.

The porous TE results in a thickened boundary layer (as shown by the authors), which effectively alters the airfoil shape and likely decreases the lift-to-drag ratio. Lift and drag measurements are missing, and the explanation provided for why the aerodynamic performance is likely not degraded is unsatisfactory.

The experimental setup description lacks scientific rigor (e.g., "fluid flow is generated by an extraordinarily high pressure difference"). Instead, refer to any of the studies on this airfoil from Feldhusen-Hoffmann and co-authors that describe the setup.

Data analysis -

1. If shock position detection was improved using certain methods, provide a time sample of shock position before and after this enhancement.
2. "The PIV data provide streamwise and streamwise orthogonal velocity information while the BOS data contain density variations." - Show data samples.
3. Why are the reported shock position and frequency different from those in Ref. 19?

Reviewer #2 (Remarks to the Author):

See the attached file.

Review NCOMMS-24-06991:

Extending the aircraft flight envelope by mitigating transonic airfoil buffet

E. Lagemann S. L. Brunton W. Schröder C. Lagemann

The manuscript introduces a proof-of-concept utilising porous materials to mitigate shock-wave buffet in the transonic flow around an airfoil. What distinguishes this approach is the placement of the porous surface not in close proximity to the shock, but at the trailing edge of the airfoil. This positioning allows for interaction with shock dynamics via the pressure field and the resulting limited boundary-layer breathing (§2.5). The study reveals a noticeable reduction in shock oscillations for two distinct internal architectures of the porous media, indicating promising prospects for the proposed control principle (see Figure 4). The authors assert that the overall aerodynamic performance remains unaffected by the presence of the porous trailing edge (§2.4). This assertion is supported by the design decision to include an impermeable layer in the airfoil mid-plane and a perforated plate at the surface. An analysis of velocity measurements in the flow (§2.4) is used to justify this claim. In future endeavours, the authors plan to employ high-fidelity simulations to bolster their assertions and utilize machine-learning techniques to gain deeper insights into the underlying physical mechanisms (§3).

The manuscript demonstrates clear and effective writing with an appropriate language level. Figures are clear, well integrated in the text, and instrumental to clarify key aspects of the study. The experimental methods are thoroughly documented and referenced, as are the data-processing techniques employed. The literature review is comprehensive, providing a solid foundation for understanding the context of the presented study and its positioning within the broader research landscape.

My main concern remains the claim of maintaining the aerodynamic performance of the airfoil, especially given the reliance on limited data. For the drag, the authors articulate an analysis based on the boundary-layer characteristics available thanks to the deployed PIV measurements (turbulent drag) and on the average shock-location (shock drag). For the lift, they rely only on the presence of the impermeable plate in the porous media. The author are well aware of this critical aspect, as they specify that “[t]o ultimately prove the effectiveness of these design aspects, complementary measurements of lift and drag forces based on a high-precision force balance are required. Unfortunately, these measurements were not feasible [...]” (lines 215–217). To overcome the lack of data, they suggest that future “complementary high-fidelity numerical simulations could provide additional aerodynamic quantities like lift and drag, which cannot be measured experimentally in the wind tunnel facility” (lines 213–214). This line of inquiry could prove to be highly challenging from a modelling and computational standpoint, primarily due to turbulence and the presence of a porous surface. To address the lift – which is, in my opinion, the *essential* aerodynamic performance not to degrade – pressure measurements could have been deployed by localised pressure transducers at the surface of the airfoil. Therefore, a good estimation of the lift could have been provided by the chord-wise integration of the pressure profiles. Since these measurements do not appear to be available in the current dataset, the authors may consider utilising the PIV measurements to reconstruct the pressure field and consequently estimate the lift (e.g., Auteri et al., 2015¹, for incompressible flows).

This aspect dims the clear outcome of attenuating the shock-wave oscillations related to buffet, achieved not through a direct interaction of the shock with the porous surface but by positioning it at the trailing edge (with the additional benefits related to noise reduction). Because of these reasons, I suggest that the submitted manuscript should be considered for publication in Nature Communications, after the above-mentioned issues have been addressed by the authors.

Minor comments and typos

lines 24–27: The authors may want also to consider aeroelastic constraints (e.g. flutter).

¹F. Auteri et al. (2015). “A novel approach for reconstructing pressure from PIV velocity measurements”. *Experiments in Fluids* 56.2. DOI: 10.1007/s00348-015-1912-z.

- line 30, 33, 34:** The Mach number, Reynolds number, and angle of attack are not defined in the introduction. Given that the authors refer to them in the initial part of the introduction, it would be beneficial for the accessibility of the article to define these terms, particularly for readers not familiar with aeronautical terminology.
- line 152:** The authors may want to define a non-dimensional frequency (i.e. a Strouhal number), since the reported displacements are already normalised by the airfoil chord. These would apply also to the figures, with a double frequency axis.
- line 179:** The authors should specify the location at which the boundary-layer quantities are extracted.
- line 257:** The authors should specify the extension of the recirculation region, i.e. the locations of the separation and reattachment points.

REVIEW

Towards extending the aircraft flight envelope by mitigating transonic airfoil buffet

Esther Lagemann^{*†1,2}, Steven L. Brunton¹, Wolfgang Schröder², and Christian Lagemann^{†1,2}

¹AI Institute in Dynamic Systems, Department of Mechanical Engineering, University of Washington, Seattle, WA 98195, United States

²Institute of Aerodynamics, RWTH Aachen University, Wüllnerstraße 5a, 52062 Aachen, Germany

^{*}corresponding author: elage@uw.edu

[†] Authors contributed equally.

We would like to thank the referees for their thorough and thoughtful remarks, which have substantially improved the quality and significance of our manuscript. In the following, we will provide detailed answers to the comments and/or questions, which are written in bold. Changes in the revised manuscript are marked in blue. References appearing in the review refer to the figures, tables, equations, sections, and page numbers of the revised manuscript, which includes all changes, if not stated otherwise.

Before providing a point-by-point response, we summarize the major changes applied to our contribution.

- In response to remarks by both referees with respect to the aerodynamic performance, which we only discussed theoretically and did not measure quantitatively, we have calculated the lift and drag contributions by deriving the pressure field from our measurement data. These calculations have been validated on results from a previous study. Thus, this new quantitative aerodynamic force analysis substantially changes the aerodynamic performance section, which also impacts the *Discussion* and extended *Methods* sections. Further details on the calculations and validations based on surface pressure measurements are described in a newly added *Supplementary Information*.
- Another major concern was related to transonic aeroelasticity. Based on this, we realize that we should have emphasized more clearly that the structural instabilities are the actual threat to flight safety. Therefore, we augmented the introduction and figure 1 in this respect. We further explain that recent research provides evidence for a direct connection between transonic aerodynamic instability, i.e., buffet, and transonic aeroelastic instability. Thus, by mitigating the fluid unsteadiness, structural instabilities should be tackled as well. We agree with referee #1 that it is of fundamental importance for the significance of our study. Therefore, we provide an extended literature review on this topic in the *Supplementary Information* for the interested reader to clarify this important aspect.
- We noticed that the coloring of PTE1 and PTE2 was reversed between the experimental setup in figure 2 and the data-related figures, which we corrected.
- We added information that was previously provided in table 2 to table 1 so that we could include an additional figure related to the aerodynamic performance without violating editorial guidelines.

Referee #1

1. **The authors link the reduction in shock buffet oscillations to the possible expansion of the flight envelope. Although aerodynamic forces due to shock buffet might induce structural loads that prohibit flight at buffet conditions, the flight envelope at high dynamic pressures is constrained by flutter, not buffet. Figure 1 is, therefore, misleading in relating buffet reduction to expanding the flight envelope. Furthermore, the study focuses on an airfoil buffet, a distinct mechanism from a wing buffet. Therefore, the assertion that the suggested TE devices can expand the flight envelope lacks substantiation.**

We agree that it has to be emphasized more clearly that the aeroelastic effects are the actual threat to the flight safety and responsible for limiting the flight envelope. Therefore, we added a paragraph to the introduction focusing on this aspect (pages 1-2, lines 29-40, 57-58, 60-61), adjusted figure 1 accordingly, and relaxed the title of the paper. Nevertheless, recent research shows that transonic aeroelastic problems arise when the flow is close to or within an unsteady state, i.e., buffet onset/offset or fully developed buffet. Thus, such non-classical transonic flutter problems possess different instability characteristics that also depend on the Mach number and the angle of attack. Therefore, mitigating the aerodynamic instability by damping transonic buffet is a very promising way to also mitigate structural instabilities in the transonic regime. We added a new section to the *Supplementary Information* (pages 10-13) that summarizes recent research with respect to transonic aeroelasticity and its relation to transonic buffet. We agree that it is important to direct the interested reader to these studies although they might be too detailed for the broad audience that the main text is dedicated to. Therefore, we compactly outline this connection between structural and aerodynamic instability in the introduction.

Moreover, we certainly agree that our study is not comprehensive in the sense that we focus on the aerodynamics of 2D buffet and do not take into account any 3D effects, aeroelastic effects, and other interactions that might influence the nature of the buffet phenomenon. Therefore, we added a paragraph to the discussion (pages 14-15, lines 369-376) to clarify this aspect. However, although transonic buffet on straight and mildly swept wings possesses 3D effects, there is evidence that the dominant aerodynamic phenomena are similar to the 2D airfoil buffet [1, 2, 3]. For example, Poplinger & Raveh [1] showed that transonic buffet on the *Benchmark Supercritical Wing*, which is a low-aspect ratio straight wing, is predominantly 2D with respect to frequency content, shock wave structure, and, likely, physical mechanism. Similarly, an analysis of a semi-infinite straight wing with an RA16SC1 section by Iovnovich & Raveh [2] revealed buffet characteristics similar to the respective 2D airfoil features, although some 3D structures related to the three-dimensionality of the turbulent boundary layer were detected as well. An experimental study by D'Aguanno et al. [3] revealed that the extent of the shock oscillation reduces for a 3D straight finite wing compared to the 2D airfoil, but the oscillation frequency is similar to the 2D buffet frequency. Moreover, buffet and buffeting onset appear when the separated region spreads from the shock wave to the trailing edge like in the 2D configuration. For a moderate sweep, the 3D buffet is characterized by a broadband bump in the frequency spectrum rather than a single buffet frequency like for the majority of 2D airfoils [4]. However, at constant spanwise positions, the shock oscillation still behaves similar to 2D buffet [4].

A more realistic high-aspect-ratio swept transonic wing will experience stronger 3D effects compared to the straight and mildly swept wings that have been primarily studied in the past. Additional phenomena like interactions with the engine nacelle [5] or the aircraft fuselage [6] cause further alterations of the transonic buffet flow. All these effects have to be considered and investigated in further detail in future, especially in the context of the efficiency of buffet mitigation strategies. Nevertheless, we think that our current study is an interesting starting point to tackle the aerodynamic unsteadiness associated

with transonic buffet and fits very well the current state of research. We will certainly pursue more complex setups in future research, starting with a swept wing to incorporate 3D effects. More avenues are outlined in the discussion (pages 14-15).

2. **The porous TE results in a thickened boundary layer (as shown by the authors), which effectively alters the airfoil shape and likely decreases the lift-to-drag ratio. Lift and drag measurements are missing, and the explanation provided for why the aerodynamic performance is likely not degraded is unsatisfactory.**

Thank you for raising this important point. To substantiate our analysis, we derived the pressure field from our PIV and BOS measurement data and used it to calculate the lift and drag contributions. Based on the reference configuration, i.e., an ordinary trailing edge, and a slightly lower Mach number $Ma = 0.73$, we were able to verify the derived pressure distribution at the wall and the resulting aerodynamic forces with previously conducted surface pressure measurements [7]. We describe the entire procedure and the validation in the (newly added) *Supplementary Information* (pages 3-10), whereas a short version is provided in the *Methods* section (pages 17-19, lines 466-515). The alteration of the lift and drag contributions in the presence of porous trailing edges is discussed in the main text (pages 10-12, lines 236-289) and substantiates our previous (theoretical) discussion of the aerodynamic performance on a quantitative level. In essence, the results show that deploying PTE1 has an adverse effect on both, lift and drag, whereas PTE2 influences the aerodynamic behavior only slightly but favorably. This diverging behavior can be traced back to the different nature of the porous materials, the influence of which also manifests in our previous observations regarding the varying impact on the boundary layer properties. Further details are described on pages 10-12, lines 236-289.

3. **The experimental setup description lacks scientific rigor (e.g., "fluid flow is generated by an extraordinarily high pressure difference"). Instead, refer to any of the studies on this airfoil from Feldhusen-Hoffmann and co-authors that describe the setup.**

Thank you for bringing this aspect to our attention. We changed the respective description and added a previous study for reference as follows (page 15, lines 402-404):

The wind tunnel itself is an intermittently working vacuum storage facility in which the fluid flow is generated by the pressure difference between a vacuum tank and an air-filled balloon [20]. Prior to each measurement run, the reservoir tank downstream of the measurement section is evacuated, while the dehumidified air is stored in the balloon upstream of the measurement section acting as a stagnation chamber.

4. Data analysis

- (a) **If shock position detection was improved using certain methods, provide a time sample of shock position before and after this enhancement.**

Thanks for pointing that out. We added an instantaneous snapshot of the shock before and after applying the 2D NA-MEMD and also provide an extract of the time-varying shock position obtained from both versions. The figures and related text are given in the *Supplementary Information* on pages 2-3 and in figure S3 and evidence the beneficial effect of using the 2D NA-MEMD.

- (b) **"The PIV data provide streamwise and streamwise orthogonal velocity information while the BOS data contain density variations." - Show data samples.**

We added examples for both measurement techniques in the *Supplementary Information* on pages 1-2. Figure S1 provides examples of the raw data and figure S2 shows instantaneous velocity and density gradient fields.

(c) **Why are the reported shock position and frequency different from those in Ref. 19?**

The previously conducted studies by Feldhusen et al. used a lower Mach number of $Ma = 0.73$. This results in a time-averaged shock wave position further downstream compared to the present study. The slightly different buffet frequency can presumably be traced back to the different measurement techniques used to detect the shock wave movement. Former studies used pressure sensors on the airfoil surface, while we relied on BOS measurements. Thus, the previous studies measured the movement of the shock foot, while we take the entire shock profile into account. We added a short description to the main paper on page 6, lines 164-170 and 174-176, to clarify this difference, which reads:

The corresponding normalized power spectrum is shown in figure 3 (b) indicating a buffet frequency of $f \approx 180$ Hz or, when expressed in non-dimensional form, a Strouhal number of $St = fc/u_\infty \approx 0.109$ based on the chord length and the freestream velocity. This value is similar to previous investigations [20,28,65] with $St \approx 0.108$. The minor deviation can be explained by the different measurement techniques used to detect the shock wave movement. These former investigations used surface pressure probes, i.e., they considered the dynamics of the shock foot, while the present setup captures the entire shock wave using BOS measurements.

[...]

Please note that the previously mentioned studies [20,28,65] were conducted at a lower Mach number of $Ma = 0.73$. Therefore, the reported time-averaged shock wave position is located further downstream compared to the present case.

Referee #2

1. **My main concern remains the claim of maintaining the aerodynamic performance of the airfoil, especially given the reliance on limited data. For the drag, the authors articulate an analysis based on the boundary layer characteristics available thanks to the deployed PIV measurements (turbulent drag) and on the average shock-location (shock drag). For the lift, they rely only on the presence of the impermeable plate in the porous media. The authors are well aware of this critical aspect, as they specify that “[t]o ultimately prove the effectiveness of these design aspects, complementary measurements of lift and drag forces based on a high precision force balance are required. Unfortunately, these measurements were not feasible [. . .]” (lines 215–217). To overcome the lack of data, they suggest that future “complementary high-fidelity numerical simulations could provide additional aerodynamic quantities like lift and drag, which cannot be measured experimentally in the wind tunnel facility” (lines 213–214). This line of inquiry could prove to be highly challenging from a modelling and computational standpoint, primarily due to turbulence and the presence of a porous surface. To address the lift – which is, in my opinion, the essential aerodynamic performance not to degrade – pressure measurements could have been deployed by localised pressure transducers at the surface of the airfoil. Therefore, a good estimation of the lift could have been provided by the chord-wise integration of the pressure profiles. Since these measurements do not appear to be available in the current dataset, the authors may consider utilising the PIV measurements to reconstruct the pressure field and consequently estimate the lift (e.g., Auteri et al., 2015, for incompressible flows).**

Many thanks for raising this concern. Following your suggestion, we derived the pressure field from our PIV and BOS measurement data and used it to calculate the lift and drag contributions. Based on the reference configuration, i.e., an ordinary trailing edge, and a slightly lower Mach number $Ma = 0.73$, we were able to verify the derived pressure distribution at the wall and the resulting aerodynamic forces with previously conducted surface pressure measurements [7]. We describe the entire procedure and the validation in the (newly added) *Supplementary Information* (pages 3-10), whereas a short version is provided in the *Methods* section (pages 17-19, lines 466-515). The alteration of the lift and drag contributions in the presence of porous trailing edges is discussed in the main text (pages 10-12, lines 236-289) and substantiates our previous (theoretical) discussion of the aerodynamic performance on a quantitative level.

In essence, the results show that deploying PTE1 has an adverse effect on both, lift and drag, whereas PTE2 influences the aerodynamic behavior only slightly but favorably. This diverging behavior can be traced back to the different nature of the porous materials, the influence of which also manifests in our previous observations regarding the varying impact on the boundary layer properties. Further details are described on pages 10-12, lines 236-289.

We also adjusted the sentence with respect to the numerical simulations on page 14, lines 362-366 as follows: **Complementary high-fidelity numerical simulations could provide time-resolved lift and drag values for the entire airfoil although an adequate implementation of a porous trailing edge at these flow conditions is highly challenging. Alternatively, additional experiments with a force balance or surface pressure measurements beyond the porous material could be conducted.**

2. **lines 24–27: The authors may want also to consider aeroelastic constraints (e.g. flutter).**

Based on a similar comment by referee #1, we adjusted the introduction and figure 1 taking aeroelastic effects like flutter into account (pages 1-2, lines 29-40). We also added an extended literature review to the *Supplementary Information* (pages 10-13) focusing on the relation between transonic aeroelastic and aerodynamic instabilities.

3. **line 30, 33, 34: The Mach number, Reynolds number, and angle of attack are not defined in the introduction. Given that the authors refer to them in the initial part of the introduction, it would be beneficial for the accessibility of the article to define these terms, particularly for readers not familiar with aeronautical terminology**

Thank you for mentioning this. We added this information at the beginning of the introduction (page 2, lines 35-37, 45).

4. **line 152: The authors may want to define a non-dimensional frequency (i.e. a Strouhal number), since the reported displacements are already normalised by the airfoil chord. These would apply also to the figures, with a double frequency axis.**

We added a second axis to the respective plots in figures 3,4 with the Strouhal number $St = fc/u_\infty$ and added this information in the main text as well (page 6, lines 165-170).

5. **line 179: The authors should specify the location at which the boundary-layer quantities are extracted.**

Thank you for raising this detail which we missed to provide in the initial manuscript. The values are extracted in the trailing edge part $x/c = [0.8, 1]$. We added the information on page 7, line 203 in the main text and in the caption of table 1.

6. **line 257: The authors should specify the extension of the recirculation region, i.e. the locations of the separation and reattachment points.**

We noticed that our explanation was misleading. By "extended" recirculation region, we meant a thickening in the wall-normal direction and not a longitudinal extension. For the studied flow conditions, the shock causes a fully separated boundary layer [8]. Precisely, the shock-induced separation of the boundary layer merges with the trailing-edge separation for $Ma \geq 0.74$ [9]. Thus, the separation location is identical to the shock wave position and the recirculation is basically terminated by the trailing edge.

For clarification, we added/adjusted the following sentences in the main paper.

- page 7, lines 193-195: For the studied flow conditions, the shock-induced boundary layer separation extends over the entire area between shock wave and trailing edge [66].
- page 8, lines 211-212: The streamwise velocities (b) are shifted to smaller values, i.e., the area containing reversed flow enlarges. This means that the recirculation region is extended in the wall-normal direction.
- page 12, lines 301-302: The streamwise velocity data additionally reveal that this effect causes a thickening of the recirculation region, i.e., an increased mass flux of upstream moving fluid, which results in a thicker boundary layer (figure 5 (a)).

References

- [1] L. Poplinger and D. E. Raveh, “Shock buffet and associated fluid–structure interactions of the benchmark supercritical wing,” AIAA Journal, vol. 61, no. 6, pp. 2381–2399, 2023.
- [2] M. Iovnovich and D. E. Raveh, “Numerical study of shock buffet on three-dimensional wings,” AIAA Journal, vol. 53, no. 2, pp. 449–463, 2015.
- [3] A. D’Aguanno, F. F. Schrijer, and B. W. van Oudheusden, “Finite-wing and sweep effects on transonic buffet behavior,” AIAA Journal, vol. 60, no. 12, pp. 6715–6725, 2022.
- [4] J. Dandois, “Experimental study of transonic buffet phenomenon on a 3d swept wing,” Physics of Fluids, vol. 28, no. 1, 2016.
- [5] T. Lürkens, M. Meinke, and W. Schröder, “Impact of 2d engine nacelle flow on buffet,” CEAS Aeronautical Journal, pp. 1–13, 2024.
- [6] L. Masini, S. Timme, and A. Peace, “Analysis of a civil aircraft wing transonic shock buffet experiment,” Journal of Fluid Mechanics, vol. 884, p. A1, 2020.
- [7] A. Feldhusen, A. Hartmann, M. Klaas, and W. Schröder, “Impact of alternating trailing-edge noise on buffet flows,” in 31st AIAA Applied Aerodynamics Conference, p. 3028, 2013.
- [8] A. Hartmann, M. Klaas, and W. Schröder, “Time-resolved stereo PIV measurements of shock–boundary layer interaction on a supercritical airfoil,” Experiments in Fluids, vol. 52, pp. 591–604, 2012.
- [9] A. Hartmann, Experimental Analysis of Wave Propagation at Buffet Flows. Ph.D. thesis, Shaker, 2012.

REVIEWERS' COMMENTS

Reviewer #2 (Remarks to the Author):

see the attached pdf file

Reviewer #3 (Remarks to the Author):

The authors have made a good faith effort to respond to the questions and comments of the reviewers. And the results of the study are worthy of publication.

The main concern of this reviewer is typified by comments such as the following. See page 14, line 3 under Section 3 Discussion.

"This aerodynamic instability [buffet] has been identified as a key element of transonic aeroelastic stability problems [9,10, 11, 12]..." This statement is literally true, but misleading. It is true that the authors of [9,10,11,12] have made this claim and that one can find combinations of aerodynamic flow parameters and elastic structural parameters where both buffet and flutter (the common terms for aeroelastic instability) occur. But this is the exceptional case (to be avoided in practice) and not the usual case. It is certainly not the case that suppressing buffet will necessarily suppress flutter or even be of benefit with respect to flutter. So I would suggest that all claims that suggest suppressing buffet will necessarily or even likely improve the prospects for suppressing flutter be omitted from the paper.

Review NCOMMS-24-06991A-Z:

Towards extending the aircraft flight envelope by mitigating transonic airfoil buffet

E. Lagemann S. L. Brunton W. Schröder C. Lagemann

The authors have addressed all the issues I raised in the first round of review. I would like to thank them for their responsiveness, which acknowledges the work of the reviewers and the time they dedicated to the peer-review process.

To estimate the aerodynamic performance of the airfoil, the authors adopted my suggestion to use PIV data to reconstruct the pressure field, enabling them to calculate the aerodynamic coefficients. They took a clever approach by also integrating data from BOS to recover the density field. This, combined with the temperature field derived from PIV data, allowed them to estimate the thermodynamic pressure under the assumption of a polytropic ideal gas. The outcomes of this analysis revealed a significant difference in aerodynamic performance between the two tested porous trailing edges. This is likely to spur further research into how porous trailing edges can reduce shock oscillations caused by buffet, as well as how their internal structure affects the overall aerodynamic performance of the airfoil.

The authors also took care of the missing information regarding aeroelastic constraints on an aircraft's flight envelope. Along with a revised introduction, they have provided a survey of the current literature on the interaction between aerodynamic and aeroelastic instabilities as "Supplementary Material". In both section, they mention "*the mechanism of classical flutter, which is caused by the coupling of unstable structural modes*" (lines 32–33). This phrasing is misleading because it suggests (i) the presence of inherently unstable structural modes, and (ii) that the coupling occurs without aerodynamic influences. I would revise this statement to better explain that the classical flutter is an aeroelastic instability that rises from two structural modes that interact with each other via the flow; this allow them to couple and combine in two aeroelastic modes, with one becoming unstable at a particular flight speed, known as the *critical flutter velocity*.

It is my opinion that the submitted manuscript should be considered for publication in Nature Communications, once the the description of the flutter mechanism has been revised by the authors.

Minor comments and typos

- The authors may want to consider in the extended literature review in the "Supplementary Material" the work by Houtman & Timme (2023)¹ on the global, aeroelastic, stability analysis of a flexible aircraft.

¹J. Houtman and S. Timme (2023). "Global stability analysis of elastic aircraft in edge-of-the-envelope flow". *Journal of Fluid Mechanics* 967, A4. DOI: 10.1017/jfm.2023.413.

REVIEW 2

Towards extending the aircraft flight envelope by mitigating transonic airfoil buffet

Esther Lagemann^{*†1,2}, Steven L. Brunton¹, Wolfgang Schröder², and Christian Lagemann^{†1,2}

¹AI Institute in Dynamic Systems, Department of Mechanical Engineering, University of Washington, Seattle, WA 98195, United States

²Institute of Aerodynamics, RWTH Aachen University, Wüllnerstraße 5a, 52062 Aachen, Germany

*corresponding author: elage@uw.edu

† Authors contributed equally.

We would like to thank the referees for taking the time to provide a thorough feedback on our previous rebuttal and for accepting the manuscript with minor changes for publication.

Please find our answers to the remaining comments, which are written in bold, below.

Referee #2

- **In both sections, they mention “the mechanism of classical flutter, which is caused by the coupling of unstable structural modes” (lines 32–33). This phrasing is misleading because it suggests (i) the presence of inherently unstable structural modes, and (ii) that the coupling occurs without aerodynamic influences. I would revise this statement to better explain that the classical flutter is an aeroelastic instability that rises from two structural modes that interact with each other via the flow; this allow them to couple and combine in two aeroelastic modes, with one becoming unstable at a particular flight speed, known as the critical flutter velocity.**

Thank you very much for letting us know that we have to describe this phenomenon in more detail. We adjusted the description on page 2 as follows:

The mechanisms of classical flutter are well understood and depend on the freestream dynamic pressure [5]. Flutter arises from two structural modes interacting via the flow, which enables a structural coupling with aeroelastic instability when reaching a particular flight speed, i.e., the critical flutter velocity.

In the Supplementary Information, we provide more details on page 11 as follows:

The mechanisms of classical aeroelastic effects like flutter are well understood and depend on the freestream dynamic pressure [8]. In essence, these phenomena are governed by the interaction of elastic and inertial forces of the structure with unsteady aerodynamic forces. Since the latter substantially increase with aircraft speed, a critical flutter velocity can be defined, which indicates the onset of

aeroelastic instability. In this case, the interaction of two structural modes, which are coupled via the flow, results in a self-excited structural vibration due to a structural mode becoming unstable. Thus, such critical flutter velocities must be excluded from the flight envelope [9].

- **The authors may want to consider in the extended literature review in the "Supplementary Material" the work by Houtman & Timme (2023) on the global, aeroelastic, stability analysis of a flexible aircraft.**

Many thanks for pointing us to this study. We have included a reference on page 12 as follows:

In one of the rare numerical studies focusing on 3D aerodynamic and structural phenomena, Houtman & Timme [28] investigated how including an elastic structure impacts the 3D aerodynamics of buffet. They demonstrated the importance of a coupled eigenmode solver in comparison to a conventional pk-type flutter method for detecting all relevant coupled modes. Similar to purely aerodynamic studies [29,30], they identified the most dominant instability at the shock foot and its downstream separated boundary layer.

Referee #3

The main concern of this reviewer is typified by comments such as the following. See page 14, line3 under Section 3 Discussion. "This aerodynamic instability [buffet] has been identified as a key element of transonic aeroelastic stability problems [9,10, 11, 12]..." This statement is literally true, but misleading. It is true that the authors of [9,10,11,12] have made this claim and that one can find combinations of aerodynamic flow parameters and elastic structural parameters where both buffet and flutter (the common terms for aeroelastic instability) occur. But this is the exceptional case (to be avoided in practice) and not the usual case. It is certainly not the case that suppressing buffet will necessarily suppress flutter or even be of benefit with respect to flutter. So I would suggest that all claims that suggest suppressing buffet will necessarily or even likely improve the prospects for suppressing flutter be omitted from the paper.

We certainly understand your concern and agree that more in-depth analyses are necessary to thoroughly understand the causal relations underlying the occurrence of buffet and flutter. We have to be careful regarding premature conclusions about the efficiency of our technology with respect to aeroelastic phenomena, which have not been part of our study. Thus, all conclusions drawn so far are based on observations from the literature and this topic is still subject to extensive on-going research. In this respect, we relaxed our descriptions of the (potential) relation between aerodynamic and aeroelastic stability problems and provide some examples below. However, we do not entirely omit the possibility that suppressing buffet can have a positive impact on aeroelastic issues from the paper since there are publications - from different research groups and both, experimental and numerical studies - that provide evidence of a link between these phenomena. But we added a clear statement to the discussion that future research is required to shed further light on this topic and that especially with respect to the efficiency of porous trailing edges, prospective studies are necessary that involve aeroelastic considerations.

- page 2: [...] attenuating the aerodynamic instability associated with transonic buffet *has the potential to tackle one of the limiting aeroelastic effects* in the transonic flight regime [...]
- page 3: Since *one class of* transonic aeroelastic instabilities of the wing structure is connected to the flow field unsteadiness associated with buffet [5, 9, 10, 12], the aerodynamic stabilization is expected to positively impact the structural behavior *for particular flow conditions*.

- page 9: *Several recent studies observed a direct connection between this aerodynamic instability and a particular class of transonic aeroelastic stability problems [9,10,11,12] (a review of recent literature is provided in the Supplementary Information), which constitute a hazard to flight safety due to their undesirable impact on the flight control system and the structural integrity. Therefore, the present findings are important for the aviation sector in outlining a promising strategy to mitigate certain safety-critical aerodynamic and aeroelastic instabilities. This could allow to safely extend the high-speed limit of the flight envelope while simultaneously reducing aircraft noise.*
- page 10: *Especially with respect to the efficiency of porous trailing edges, prospective studies should involve aeroelastic considerations to contribute to the on-going research question to which extent transonic buffet and structural stability issues are related.*